# Beyond Zero Initialization: Investigating the Impact of Non-Zero Initialization on LoRA Fine-Tuning Dynamics

Shiwei Li [1 2 *]  Xiandi Luo [1 *]  Xing Tang [2]  Haozhao Wang [1]  Hao Chen [3]  Weihong Luo [3]  Yuhua Li [1]
Xiuqiang He [2]  Ruixuan Li [1]

## Abstract

Low-rank adaptation (LoRA) is a widely used parameter-efficient fine-tuning method. In standard LoRA layers, one of the matrices, $A$ or $B$, is initialized to zero, ensuring that fine-tuning starts from the pretrained model. However, there is no theoretical support for this practice. In this paper, we investigate the impact of non-zero initialization on LoRA's fine-tuning dynamics from an infinite-width perspective. Our analysis reveals that, compared to zero initialization, simultaneously initializing $A$ and $B$ to non-zero values improves LoRA's robustness to suboptimal learning rates, particularly smaller ones. Further analysis indicates that although the non-zero initialization of $AB$ introduces random noise into the pretrained weight, it generally does not affect fine-tuning performance. In other words, fine-tuning does not need to strictly start from the pretrained model. The validity of our findings is confirmed through extensive experiments across various models and datasets. The code is available at https://github.com/Leopold1423/non_zero_lora-icml25.

## 1. Introduction

Large language models (LLMs), such as GPT-4 (OpenAI, 2024) and DeepSeek-V3 (DeepSeek-AI, 2024), have achieved unprecedented performance across a broad range of tasks. These state-of-the-art models typically contain hundreds of billions of parameters. Consequently, fully fine-tuning these models is computationally expensive and

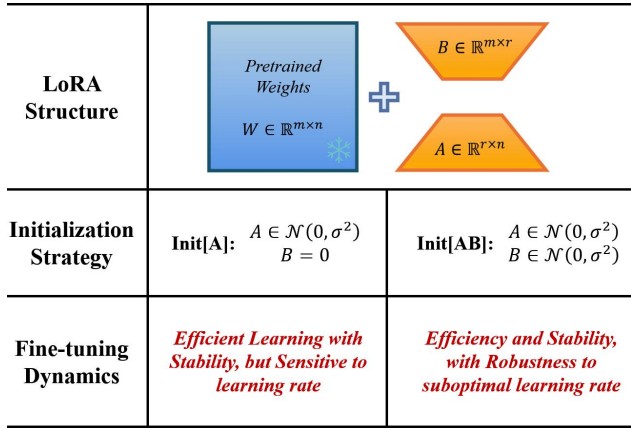

| | | |
|---|---|---|
| **LoRA Structure** | *Pretrained Weights* $W \in \mathbb{R}^{m \times n}$ | $B \in \mathbb{R}^{m \times r}$ $A \in \mathbb{R}^{r \times n}$ |
| **Initialization Strategy** | **Init[A]:** $A \in \mathcal{N}(0, \sigma^2)$ $B = 0$ | **Init[AB]:** $A \in \mathcal{N}(0, \sigma^2)$ $B \in \mathcal{N}(0, \sigma^2)$ |
| **Fine-tuning Dynamics** | *Efficient Learning with Stability, but Sensitive to learning rate* | *Efficiency and Stability, with Robustness to suboptimal learning rate* |

*Figure 1.* LoRA's fine-tuning dynamics with zero initlization (i.e., `Init[A]`) and non-zero initialization (i.e., `Init[AB]`). Compared to `Init[A]`, `Init[AB]` improves LoRA's robustness to suboptimal learning rates, generally leading to better performance.

storage-intensive, presenting significant challenges for their practical deployment. To address these issues, various parameter-efficient fine-tuning (PEFT) methods have been proposed to reduce the number of trainable parameters, thereby decreasing the cost of fine-tuning (Xu et al., 2023).

Among these PEFT methods, low-rank adaptation (LoRA) (Hu et al., 2022) has gained considerable attention due to its efficiency and minimal impact on inference latency. LoRA approximates the update of the pretrained weight matrix $W$ using smaller low-rank matrices, $A$ and $B$, as illustrated in Figure 1. The forward pass of LoRA is formulated as $Y = (W + \alpha/r BA)X$, where $r$ is the LoRA rank and $\alpha$ is a tunable scaling factor. During fine-tuning, $W$ remains fixed, while only matrices $A$ and $B$ are updated, usually via the Adam optimizer (Kingma & Ba, 2015).

In standard LoRA layers, either $A$ or $B$ is initialized to zero, ensuring that fine-tuning starts from the pretrained model. Building on this premise, Hayou et al. (2024b) explores the difference between applying Kaiming initialization (He et al., 2016) to $A$ and $B$. However, there is no theoretical justification for the zero initialization of LoRA, which raises

---

*Equal contribution. [1]Huazhong University of Science and Technology, Wuhan, China [2]Shenzhen Technology University, Shenzhen, China [3]FiT, Tencent, Shenzhen, China. Correspondence to: Xing Tang <xing.tang@hotmail.com>, Haozhao Wang <hz_wang@hust.edu.cn>, Ruixuan Li <rxli@hust.edu.cn>.

*Proceedings of the 42nd International Conference on Machine Learning*, Vancouver, Canada. PMLR 267, 2025. Copyright 2025 by the author(s).

the question: **Is it optimal to initialize $A$ or $B$ to zero?** To answer this question, we explore the impact of initialization on LoRA's fine-tuning dynamics. Let `Init[A]` denote the case where $A$ is randomly initialized and $B$ is set to zero, while `Init[AB]` denotes the case where both $A$ and $B$ are initialized to non-zero values, as illustrated in Figure 1. Our analysis demonstrates that, compared to `Init[A]`, `Init[AB]` improves LoRA's robustness to suboptimal learning rates, particularly smaller ones. Notably, learning rate decay is commonly applied during fine-tuning, making smaller learning rates more prevalent.

However, non-zero initialization introduces noise to the pretrained model before fine-tuning, which raises another question: **Is it necessary for fine-tuning to strictly start from the pretrained model?** Note that pretrained weights are usually suboptimal for downstream tasks, suggesting that the pretrained model itself contains inherent noise. Thus, we conjecture that the noise introduced by non-zero initialization can also be corrected through fine-tuning, without additional optimization cost. Extensive verification shows that, with appropriate initialization variance, the noise introduced by non-zero initialization does not degrade the fine-tuning accuracy. Notably, the range of suitable initialization variances is quite broad, encompassing that used in Kaiming initialization. In conclusion, fine-tuning does not have to strictly start from the pretrained model.

The main contributions of this paper can be summarized as two novel insights that challenge the traditional inertial constraints of LoRA fine-tuning:

1. You do not need to initialize LoRA's matrices $A$ or $B$ to zero. Compared to zero initialization, non-zero initialization enhances LoRA's robustness to the learning rate and generally achieves better fine-tuning accuracy.

2. You do not need to start fine-tuning strictly from a pretrained model. Although non-zero initialization alters the starting point of LoRA fine-tuning, it does not impact the fine-tuning performance, as long as an appropriate initialization variance is employed.

3. Experimental results confirm the validity of our theory and findings. LoRA with non-zero initialization consistently achieves superior accuracy compared to zero initialization, particularly at smaller learning rates.

## 2. Preliminaries

### 2.1. Scaling Theory of Neural Networks

The scaling theory of neural networks is widely used to guide the selection of hyperparameters, such as learning rates (Hayou et al., 2024a), initialization strategies (He et al., 2016; Yang, 2019; Hayou et al., 2024b) and depth

parametrizations (Bordelon et al., 2024; Yang, 2020) etc. It typically involves analyzing the statistical properties of key quantities, such as pre-activation values, by extending the network's width or depth to infinity. Subsequently, this analysis is leveraged to refine hyperparameters, ensuring favorable convergence properties under these extreme conditions. For instance, Kaiming initialization (He et al., 2016) suggests that the variance of the weights in hidden layers should be scaled by $1/n$ to prevent excessively large preactivations as the model width $n$ increases. More details on the scaling theory are provided in Appendix A.1.

In this paper, we apply the scaling theory to investigate the fine-tuning dynamics of LoRA from the perspective of infinite width.[1] Specifically, we explore how to set the learning rate and initialization strategy for LoRA to achieve desired performance. In our analysis, we assume that only the model width $n$ increases, while other factors, such as the LoRA rank, number of layers, sequence length, and number of training steps, remain constant. To characterize the asymptotic behavior of the analyzed variables as $n$ increases, we further introduce the following notations, which have also been utilized in previous analysis of LoRA, particularly in the context of learning rates (Hayou et al., 2024a) and zero-initialization strategies (Hayou et al., 2024b).

**Notations.** Let $c_n \in \mathbb{R}$ and $d_n \in \mathbb{R}^+$ be sequences. If there exists a constant $\kappa > 0$ such that $c_n < \kappa d_n$, we write $c_n = \mathcal{O}(d_n)$. If $c_n > \kappa d_n$, we write $c_n = \Omega(d_n)$. If both $c_n = \mathcal{O}(d_n)$ and $c_n = \Omega(d_n)$ hold simultaneously, we write $c_n = \Theta(d_n)$. For a vector sequence $c_n = (c_n^i)_{1 \le i \le k} \in \mathbb{R}^k$ ($k > 0$), we write $c_n = \mathcal{O}(d_n)$ if and only if $c_n^i = \mathcal{O}(d_n)$ for all $i \in [k]$. This similarly applies to $\Theta$ and $\Omega$. Additionally, when the sequence $c_n$ is a vector of random variables, convergence is understood to be convergence of its $\ell_2$ norm. To track the exponent of the asymptotic behaviour of some quantity $v$, we further introduce the $\gamma$ operator, defined as $v = \Theta(n^{\gamma[v]})$. For example, $\gamma[n] = 1$, $\gamma[1] = 0$, and $\gamma[0] = -\infty$.[2] For two real-valued variables $u$ and $v$, we have $\gamma[v \times u] = \gamma[v] + \gamma[u]$, and $\gamma[v + u] = \max\{\gamma[v], \gamma[u]\}$ provided that $v \ne -u$. More details about the $\gamma$-operator are provided in Appendix A.2.

Next, we focus on the asymptotic behavior of the learning rate, $\gamma[\eta]$, as the network width $n \to \infty$, rather than its exact value. Given $\gamma[\eta]$, $\eta = \Theta(n^{\gamma[\eta]}) \approx c \cdot n^{\gamma[\eta]}$, where $c > 0$ is a constant and lower-order terms are neglected. As $n \to \infty$, the term $n^{\gamma[\eta]}$ dominates, making $\gamma[\eta]$ the key factor determining the asymptotic behavior of $\eta$. While the constant $c$ is important for exact values, it doesn't influence the asymptotic scaling behavior. Ignoring the influence of $c$, perturbations to $\gamma[\eta]$ and $\eta$ are effectively equivalent. Therefore, we analyze the impact of perturbations in $\gamma[\eta]$

---

[1]Note that the width $n$ of LLMs is typically large ($n > 10^3$).
[2]$\gamma[0] = -\infty$ is written as a limit of $\gamma[n^\beta]$ when $\beta \to -\infty$.

on fine-tuning dynamics, excluding $c$ from consideration. This analysis aims to guide the selection of learning rates and initialization strategies based on asymptotic properties, rather than computing precise values that depend on $c$.

## 2.2. LoRA and LoRA Features

Based on the previous analysis of LoRA presented in (Hayou et al., 2024a;b), we adopt the same notation and consider a general neural network of the following form:

$$\begin{cases} Y_{in}(X) = W_{in}X, \\ Y_l(X) = \mathcal{F}_l(W_l, Y_{l-1}(X)),\ l \in [L], \\ Y_{out}(X) = W_{out}Y_L(X), \end{cases} \quad (1)$$

where $X \in \mathbb{R}^d$ denotes the input vector, and $L \geq 1$ is the network depth. Let $n$ denote the network width and $\mathcal{F}$ denote the nonlinear mapping function. $W_l \in \mathbb{R}^{n \times n}$ is the weight matrix of the $l$-th hidden layer. $W_{in}$ and $W_{out}$ correspond to the weight matrices of the input and output layers, respectively. This model represents a pretrained neural network that will be fine-tuned for a specific downstream task using the LoRA framework defined below.

**Definition 1** (LoRA). *Let $W \in \mathbb{R}^{n_1 \times n_2}$ be a weight matrix from the pretrained model. LoRA approximates the update of $W$ using low-rank matrices, such that the value of the pretrained weight becomes $W + \alpha/r BA$. Here, only $B \in \mathbb{R}^{n_1 \times r}$ and $A \in \mathbb{R}^{r \times n_2}$ are trainable. The rank $r \ll \min(n_1, n_2)$, and $\alpha \in \mathbb{R}$ is a tunable scaling factor.*

For each LoRA layer within the network, let $\underline{Z}$ denote its input and let $\bar{Z}$ denote the output after incorporating the pretrained weights, i.e., $\bar{Z} = W\underline{Z} + \alpha/r BA\underline{Z}$. In this paper, we primarily investigate the statistical properties of the output of the LoRA module, i.e., $BA\underline{Z}$. To this end, we further introduce the LoRA features as follows:

**Definition 2** (LoRA Features). *Given a general neural architecture and a LoRA (Definition 1) layer, we define the LoRA features $(Z_A, Z_B)$ as $Z_A = A\underline{Z}$ and $Z_B = BZ_A = BA\underline{Z}$. The superscript $t$ of $Z_A^t, Z_B^t$ represents the value of the LoRA features in the $t$-th step of fine-tuning.*

Definitions 1 and 2 are straightforward and have been used in (Hayou et al., 2024a;b). Next, we focus on the training dynamics of the final output $Z_B$ as $n$ approaches infinity.

## 3. Fine-Tuning Dynamics of LoRA with Adam

In this section, we first introduce two critical statistics of LoRA: stability and efficiency, as discussed in Section 3.2, along with their respective solution sets for the learning rate and initialization. Next, we investigate the sensitivity of these solutions to the learning rate in Section 3.3 and explain how non-zero initialization improves their robustness.

Additionally, a simple toy model is presented in Section 3.4 to validate our theoretical findings. Finally, we explore whether non-zero initialization affects fine-tuning accuracy by deviating from the intended purpose of zero initialization, which is to start fine-tuning from a pretrained model.

## 3.1. Simplified Settings and the Optimizer

To simplify the analysis, we adopt the setup proposed in (Hayou et al., 2024a;b). Specifically, we assume that the fine-tuning dataset comprises a single data point, $(x, y)$.[3] The optimization goal is to minimize the loss calculated using the pretrained model that incorporates LoRA.

Moreover, we train only one LoRA layer to isolate the contribution of individual LoRA layers to feature learning. Let $\underline{Z}$ denote the input to the LoRA layer computed from $x$, and let $d\bar{Z}^t$ represent the gradient of the loss function, evaluated at the data point $(x, y)$, with respect to $\bar{Z}$. During fine-tuning, the input $\underline{Z}$ remains constant as the step $t$ increases. In contrast, $d\bar{Z}^t$ varies with $t$, as $\bar{Z}^t$ evolves over time. Next, we examine the behavior of $Z_B$ (Definition 2) as $n$ approaches infinity from three perspectives: stability, efficiency, and robustness. Formal definitions of these perspectives will be provided in subsequent sections.

Adam (Kingma & Ba, 2015) is a commonly used optimizer for fine-tuning LLMs. Therefore, we focus on the fine-tuning dynamics of LoRA with the Adam optimizer. A similar analysis for the SGD optimizer (Bottou, 2010) is provided in Appendix C, where we observe that Adam outperforms SGD in enhancing LoRA's feature learning capabilities, primarily due to the gradient normalization process. The conclusions drawn in this section are partially applicable to SGD; further details can be found in Appendix C.

## 3.2. Stability and Efficiency of LoRA

To ensure that the LoRA layers produce stable outputs, it is expected that $Z_B$ remains within a reasonable range throughout the fine-tuning process. Consequently, the concept of stability is introduced and defined as follows:

**Definition 3** (Stability). *A LoRA fine-tuning process is considered stable if, for all LoRA layers and for all $t > 1$,[4] $Z_B^t = \Theta(1)$ as the width $n$ approaches infinity.*

Stability indicates that as the width increases, the output of LoRA, $Z_B$, remains bounded and does not diminish, demonstrating LoRA's effective and positive contribution to fine-tuning outcomes. For the sake of analytical clarity,

---

[3]This simplification is made for analytical purposes only; the analysis can be extended to mini-batched gradients without changing the conclusions. However, additional assumptions are required to ensure the rigor of such an extension.

[4]The case where $t = 1$ is excluded because, in standard LoRA, $B$ is typically initialized to 0, resulting in $Z_B^1 = 0$.

we do not consider LoRA's internal stability (i.e., $Z_A$) in this section. Instead, a detailed analysis and discussion of internal stability are provided in Appendix B.2.

In addition to the stability of $Z_B$, we are more concerned about whether $Z_B$ is effectively updated during fine-tuning. At the $t$-th step, the update to $Z_B$ is given by:

$$\Delta Z_B^t = \underbrace{B_{t-1}\Delta Z_A^t}_{\delta_t^1} + \underbrace{\Delta B_t Z_A^{t-1}}_{\delta_t^2} + \underbrace{\Delta B_t \Delta Z_A^t}_{\delta_t^3}, \quad (2)$$

As discussed in (Hayou et al., 2024a;b), the terms $\delta_t^1$ and $\delta_t^2$ denote 'linear' updates, reflecting the effects of training $A$ and $B$ independently. The third term, $\delta_t^3$, represents a 'multiplicative' update, capturing the combined effect of optimizing both $A$ and $B$ simultaneously. Ideally, as $n$ approaches infinity, $\Delta Z_B$ should remain $\Theta(1)$ to ensure effective updates to $Z_B$ during fine-tuning. A similar condition was proposed in (Yang, 2020) for pretraining and later adopted by (Hayou et al., 2024a;b) as a metric for efficient feature learning in LoRA. According to (Hayou et al., 2024a;b), it is essential for both $\delta_t^1$ and $\delta_t^2$ to be $\Theta(1)$ to ensure that the updates of $A$ and $B$ contribute positively to $\Delta Z_B$.[5] Based on the above analysis, the efficiency of LoRA's feature learning is defined as follows:

**Definition 4** (Efficiency). *A LoRA fine-tuning process is considered efficient if it is stable (Definition 3) and, for all LoRA layers and for all $t > 1$, $\delta_t^1 = \Theta(1)$ and $\delta_t^2 = \Theta(1)$.*

In contrast to stability, efficiency emphasizes the effective updating of $Z_B$. We now analyze the effects of initialization and learning rate on both stability and efficiency. At the $t$-th step of fine-tuning, the LoRA weights are updated as $A_t = A_{t-1} - \eta_A g_A^{t-1} = A_0 - \eta_A \sum_{i=0}^{t-1} g_A^i$ and $B_t = B_{t-1} - \eta_B g_B^{t-1} = B_0 - \eta_B \sum_{i=0}^{t-1} g_B^i$, where $(\eta_A, \eta_B)$ are the learning rates, $(A_0, B_0)$ are the initialization values, and $(g_A, g_B)$ represent the normalized gradients generated by the Adam optimizer. Note that both $g_A$ and $g_B$ are of order $\Theta(1)$ due to the gradient normalization process. By substituting $\Delta Z_A^t = -\eta_A g_A \underline{Z}$ and $\Delta B^t = -\eta_B g_B$ into Eq.(2), and applying the $\gamma$-operator, the conditions for LoRA to achieve stability and efficiency can be expressed as:

$$\begin{cases} \gamma[\delta_t^1] = \gamma[-\eta_A B_{t-1} g_A^{t-1}\underline{Z}] = 0 & (\delta_t^1 = \Theta(1)) \\ \gamma[\delta_t^2] = \gamma[-\eta_B g_B^{t-1} A_{t-1}\underline{Z}] = 0 & (\delta_t^2 = \Theta(1)) \\ \gamma[Z_B^{t-1}] = \gamma[B_{t-1} A_{t-1}\underline{Z}] = 0 & (Z_B^{t-1} = \Theta(1)) \end{cases} \quad (3)$$

Using the properties of the $\gamma$-operator, we can express the

above equations in terms of $(A_0, B_0)$ and $(\eta_A, \eta_B)$ as:[6]

$$\begin{cases} \max(\gamma[B_0], \gamma[\eta_B]) + \gamma[\eta_A] + 1 = 0, \\ \max(\gamma[A_0], \gamma[\eta_A]) + \gamma[\eta_B] + 1 = 0, \\ \max(\gamma[A_0], \gamma[\eta_A]) + \max(\gamma[B_0], \gamma[\eta_B]) + 1 = 0, \end{cases} \quad (4)$$

which further leads to the following conditions:

$$\begin{cases} \gamma[\eta_A] + \gamma[\eta_B] = -1, \\ \gamma[A_0] \leq \eta_A, \quad \gamma[B_0] \leq \eta_B. \end{cases} \quad (5)$$

By default, a uniform learning rate is used for the weight matrices $A$ and $B$, i.e., $\eta_A = \eta_B$. Based on this condition, the solution to the above equations is

$$\begin{cases} \gamma[\eta_A] = \gamma[\eta_B] = -1/2, \\ \gamma[A_0] \leq -1/2, \quad \gamma[B_0] \leq -1/2, \end{cases} \quad (6)$$

where $\gamma[A_0]$ and $\gamma[B_0]$ cannot both be equal to $-\infty$ simultaneously, as this would imply that both $A$ and $B$ are zero, preventing optimization via gradient descent algorithms.

In standard LoRA, $A$ is initialized using Kaiming initialization, while $B$ is initialized to zero. This initialization strategy is denoted as Init[A]. In contrast, when $B$ is initialized using Kaiming initialization and $A$ is initialized to zero, the strategy is referred to as Init[B]. Let $\sigma_A^2$ and $\sigma_B^2$ represent the variances of $A_0$ and $B_0$, respectively. For Init[A], $\sigma_A^2 = 1/n$ and $\sigma_B^2 = 0$, which results in $\gamma[A_0] = -1$ and $\gamma[B_0] = -\infty$, thereby satisfying Eq.(6). However, for Init[B], $\sigma_A^2 = 0$ and $\sigma_B^2 = 1/r$, resulting in $\gamma[A_0] = -\infty$ and $\gamma[B_0] = 0$, which fails to meet the desired conditions.[7] Consequently, we conclude that LoRA achieves stable and efficient feature learning when using Adam with Init[A], but not with Init[B]. This conclusion aligns with the findings of (Hayou et al., 2024b).

Similarly, when using the SGD optimizer, solutions to ensure stability and efficiency for LoRA can be derived as:

$$\begin{cases} \gamma[\eta_A] = \gamma[\eta_B] = -1/2, \\ \gamma[A_0] \leq -3/4, \quad \gamma[B_0] \leq -1/4, \end{cases}$$

where at least one of the two inequalities must hold with equality. It is evident that neither Init[A] nor Init[B] can satisfy the desired conditions when using the SGD optimizer. Furthermore, we highlight that Adam offers more flexible initialization options than SGD, a benefit attributed to the gradient normalization process. Detailed proofs and explanations can be found in Appendix C.

---

[5]One might wonder why $\delta_t^3$ is not required to be $\Theta(1)$. This is because as long as LoRA is stable and $\delta_t^1$, $\delta_t^2$ are both $\Theta(1)$, then $\delta_t^3 = \Theta(1)$ (i.e., $\gamma[\delta_t^3] = \gamma[\eta_A] + \gamma[\eta_B] + 1 = 0$) is guaranteed, as shown in Eq.(5).

[6]A detailed proof of this conversion is given in Appendix B.1.

[7]In the case where $\eta_A = \eta_B = \eta$, Init[B] can only achieve stability when $\gamma[\eta] = -1$. LoRA+ (Hayou et al., 2024a) decouples the learning rate of matrices $A$ and $B$ (i.e., $\gamma[\eta_A] = -1$ and $\gamma[\eta_B] = 0$), thereby enabling LoRA to achieve both stability and efficiency when using Init[B].

## 3.3. Robustness of LoRA

The solution presented in Eq.(6) demonstrates that LoRA's stability and efficiency impose stricter constraints on the learning rate, while the range of feasible initialization values is comparatively broader. Note that strictly setting $\gamma[\eta_A]$ and $\gamma[\eta_B]$ to $-1/2$ is particularly challenging in practical scenarios, as the constant in $\Theta$ is generally intractable. Consequently, we investigate the impact of deviations in $\gamma[\eta]$ from $-\frac{1}{2}$ on LoRA's stability and efficiency. For the case where $\gamma[\eta] > -1/2$, $\gamma[\delta_1] = \gamma[\delta_2] = \gamma[Z_B] = 2\gamma[\eta] + 1 > 0$, which inevitably causes instability and inefficiency. Therefore, learning rate decay is commonly applied during fine-tuning to prevent excessively large learning rates. In the following, we primarily focus on the case where $\gamma[\eta] \leq -1/2$.

First, when $\gamma[\eta_A] \leq \gamma[A_0] \leq -1/2$, $\max(\gamma[A_0], \gamma[\eta_A]) = \gamma[A_0]$. In this case, $\gamma[\eta_A] < -1/2$ will not affect $\gamma[\delta_2]$ and $\gamma[Z_B]$. Similarly, when $\gamma[\eta_B] \leq \gamma[B_0] \leq -1/2$, $\gamma[\eta_B] < -1/2$ will not affect $\gamma[\delta_1]$ and $\gamma[Z_B]$. Conversely, when $\gamma[\eta_A] > \gamma[A_0]$, $\gamma[\delta_1] = \gamma[\eta_A] + \gamma[\eta_B] + 1 = 2\gamma[\eta] + 1$, which implies that changes in $\eta$ will result in a quadratic change in $\delta_1$. The same holds for $\delta_2$ and $Z_B$. Clearly, in this case where $\gamma[\eta_A] > \gamma[A_0]$ and $\gamma[\eta_B] > \gamma[B_0]$, LoRA is more sensitive to fluctuations in the learning rate. Based on these insights, we formally define the robustness of LoRA's stability and efficiency as follows:

**Definition 5** (Robustness). *Assume a LoRA fine-tuning process is stable and efficient (Definitions 3-4) with $\gamma[\eta] = \gamma^*$. The process is considered robust with respect to the learning rates $\gamma[\eta_A] \leq \gamma^*$ and $\gamma[\eta_B] \leq \gamma^*$ if, for all LoRA layers and for all $t > 1$, the following relations hold: $P(\gamma[\delta_1] \mid \gamma[\eta_B]) = P(\gamma[\delta_1])$, $P(\gamma[\delta_2] \mid \gamma[\eta_A]) = P(\gamma[\delta_2])$, and $P(\gamma[Z_B] \mid \gamma[\eta_A]) = P(\gamma[Z_B] \mid \gamma[\eta_B]) = P(\gamma[Z_B])$.*[8]

Robustness focuses on the impact of suboptimal learning rates, particularly those lower than the optimal value, on the stability and efficiency (i.e., $\delta_1, \delta_2$ and $Z_B$). Specifically, $\delta_1$ reflects the effect of training $A$ independently, and robustness seeks its independence from $\eta_B$. The same principle applies to $\delta_2$. To achieve this, both $\gamma[A_0]$ and $\gamma[B_0]$ should be set to $-1/2$,[9] which leads to the following relationships for all learning rates satisfying $\gamma[\eta_A] \leq -1/2$ and $\gamma[\eta_B] \leq -1/2$.

$$\begin{cases} \gamma[\delta_1] = \gamma[B_0] + \gamma[\eta_A] + 1 = \gamma[\eta_A] + 1/2, \\ \gamma[\delta_2] = \gamma[A_0] + \gamma[\eta_B] + 1 = \gamma[\eta_B] + 1/2, \\ \gamma[Z_B] = \gamma[A_0] + \gamma[B_0] = 1. \end{cases}$$

Clearly, $\delta_1$ and $\delta_2$ vary linearly with changes in the learning rates $\eta_A$ and $\eta_B$, respectively, while $\gamma[Z_B] = 1$ remains

---

[8]The conditional probability $P(x \mid y) = P(x)$ indicates that $x$ is independent of $y$.

[9]Setting $\gamma[A_0] = \gamma[B_0] = s < -1/2$ ensures robustness only for learning rates that satisfy $\gamma[\eta] \leq s$.

unaffected, thereby ensuring consistent stability. Based on the above analysis, we derive Theorem 1, which presents the initialization strategy that optimally supports robustness (Definition 5) while maintaining stability and efficiency (Definitions 3 and 4) for LoRA fine-tuning.

**Theorem 1** (Robust LoRA (Informal)). *A LoRA fine-tuning process optimized with Adam exhibits optimal robustness (Definition 5) to the learning rate when $A_0 = \Theta(n^{-1/2})$ and $B_0 = \Theta(n^{-1/2})$.*

This initialization scheme is referred to as `Init[AB]`, where it is observed that $A_0 B_0 = \Theta(n^{-1})$. This is equivalent to treating $AB$ as a whole and then performing Kaiming initialization, rather than directly performing Kaiming initialization for $A$ or $B$ as done in `Init[A]` or `Init[B]`. As shown in Table 1, `Init[AB]` can simultaneously achieve stability, efficiency, and robustness, whereas `Init[A]` and `Init[B]` cannot.

Table 1. A comparison between various initialization strategies.

| Initialization | Stability | Efficiency | Robustness |
|---|---|---|---|
| `Init[B]` | ✓ | ✗ | ✗ |
| `Init[A]` | ✓ | ✓ | ✗ |
| `Init[AB]` | ✓ | ✓ | ✓ |

However, enforcing $A_0$ and $B_0$ to be exactly $\Theta(n^{-1/2})$ is challenging (similar to the learning rate), as the constant within $\Theta$ is typically intractable. Therefore, we do not impose a strict requirement on $A_0$ and $B_0$ to be of order $\Theta(n^{-1/2})$. Instead, our main focus is ensuring that $A_0/B_0 = \Theta(1)$, which guarantees that $A$ and $B$ are initialized to non-zero values simultaneously. For simplicity, we ignore the constant in $\Theta(1)$, implying that the variances of $A_0$ and $B_0$ are equal. Compared to initializing $B$ to zero, this at least enhances the robustness of LoRA fine-tuning with respect to $\eta_B$. Finally, we formally define the non-zero initialization for LoRA as follows:

**Definition 6** (Non-Zero Initialization). *A LoRA layer is non-zero initialized if the weight matrices $A$ and $B$ are initialized with the same variance, i.e., $\sigma_A^2 = \sigma_B^2$.*

## 3.4. Toy Model

In this section, we verify the advantages of non-zero initialization in LoRA by fine-tuning the following toy model, which is actually the general model in Eq.(1) with $L = 1$.

$$f(X) = W_{out}\mathcal{F}((W_0 + BA)\mathcal{F}(W_{in}X)),$$

where $X \in \mathbb{R}^d$ is the input vector, $W_{in} \in \mathbb{R}^{n \times d}$, $W_{out} \in \mathbb{R}^{10 \times n}$, and $W_0 \in \mathbb{R}^{n \times n}$ are the pretrained weights. $A \in \mathbb{R}^{r \times n}$ and $B \in \mathbb{R}^{n \times r}$ are the LoRA weights. The dimensions $(d, n, r)$ are set to 784, 4096, and 32. The MNIST (Deng, 2012) and FashionMNIST (Xiao et al., 2017)

datasets are used for pretraining and fine-tuning, respectively. During fine-tuning, only matrices $A$ and $B$ are trainable, while $W_{in}, W_0$ and $W_{out}$ remain fixed. Adam is used as the optimizer. More details about the toy model can be found in Appendix D.1. Subsequently, we compare the test loss of LoRA under zero and non-zero initializations, varying learning rates and initialization variances. The results, presented in Figure 2, demonstrate that the performance of LoRA with zero initialization deteriorates significantly at lower learning rates, whereas the non-zero initialization consistently yields better performance, aligning with our theoretical expectations.

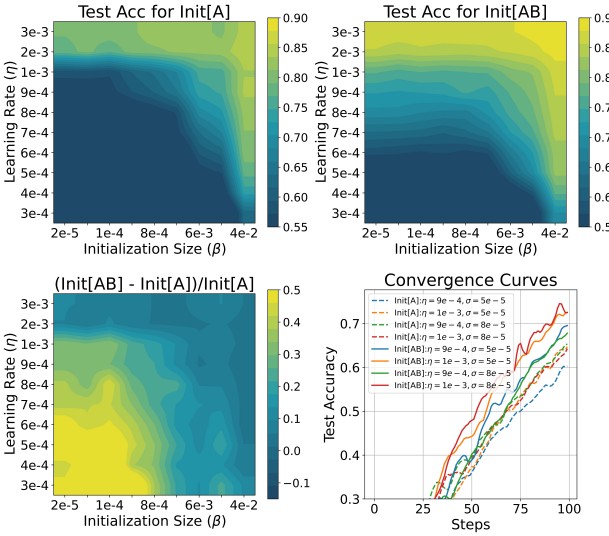

*Figure 2.* Test loss of LoRA fine-tuned with zero initlization (`Init[A]`) and non-zero initialization (`Init[AB]`). The lower-left corner presents the improvement ratio of `Init[AB]` relative to `Init[A]`, while the lower-right corner presents the convergence curves of `Init[AB]` and `Init[A]` under specific settings.

### 3.5. Starting Point of LoRA Fine-Tuning

Previous studies have unintentionally employed the non-zero initialization for LoRA (Li et al., 2024; Meng et al., 2024; Wang et al., 2024). LoftQ (Li et al., 2024) fine-tunes a quantized pretrained model and uses quantization errors to initialize the LoRA module. Specifically, it generates the initialization values for LoRA by performing truncated Singular Value Decomposition (SVD) on the quantization errors, thereby minimizing these errors in the pretrained model. PISSA (Meng et al., 2024) suggests performing SVD on the pretrained weights and initializing $A$ and $B$ using the singular vectors and largest singular values. It believes that updating the principal components of the pretrained model accelerates convergence and enhances performance. LoRA-GA (Wang et al., 2024) proposes initializing

$A$ and $B$ based on gradient information. This involves computing the gradient of the pretrained model over several steps of gradient descent and applying truncated SVD on the gradient to derive the initialization values for matrices $A$ and $B$. Although these previous methods utilize non-zero initialization for LoRA, they do not discuss the benefits nor provide a theoretical foundation for such initialization. This study bridges these gaps and offers theoretical support for non-zero initialization techniques.

Standard LoRA employs zero initialization to ensure that fine-tuning starts from the pretrained model. However, non-zero initialized LoRA introduces noise to the pretrained model before fine-tuning begins. A straightforward solution to this problem is to subtract $\frac{\alpha}{r}A_0B_0$ from the pretrained weights, as proposed by Wang et al. (2024). However, this requires modifications to the pretrained model, complicating its reuse. This challenge becomes particularly evident in multi-LoRA scenarios, where the pretrained model is shared across multiple LoRA modules for different tasks (Sheng et al., 2023). In such cases, it is required that the initialization of different LoRA modules remains consistent.

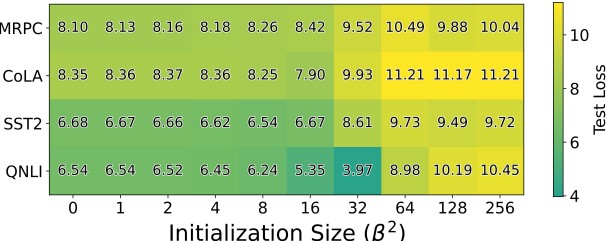

*Figure 3.* Test loss of un-fine-tuned T5-Base on different dataset with various initialization size ($\beta^2$). Note that the initialization variances of matrices $A$ and $B$ are both set to $(\beta\sigma_k)^2$, where $\sigma_k^2$ is the variances of $A$ when initialized with Kaiming initialization.

Above discussion raises the question of whether it is necessary to subtract $\frac{\alpha}{r}A_0B_0$ from the pretrained model. Note that pretrained weights are typically not optimal for downstream tasks. Therefore, we conjecture that introducing small perturbations to the pretrained weights will not significantly affect their performance on downstream tasks, nor will it substantially impact convergence during fine-tuning. To test this hypothesis, we conducted a preliminary evaluation using the T5-Base model (Raffel et al., 2020). Specifically, we compared the accuracy of the un-fine-tuned T5-Base on several datasets of the GLUE benchmark (Wang et al., 2019), utilizing non-zero initialized LoRA with various initialization variances. As shown in Figure 3, the non-zero initialization does not impact the initial performance of the pretrained model on downstream tasks unless the initialization variance is exceedingly large. Therefore, we conclude that, with appropriate initialization variances,

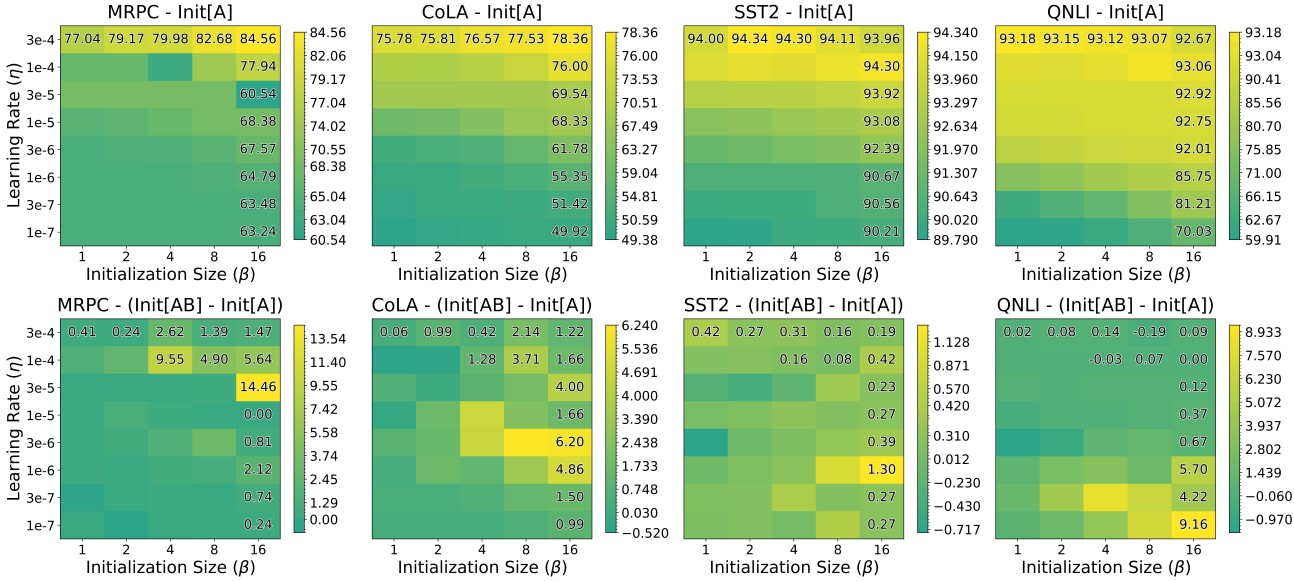

*Figure 4.* Fine-tuning accuracy of T5-Base on different datasets with various learning rates ($\eta$) and initialization sizes ($\beta$). Note that the initialization variance is set to $(\beta\sigma_k)^2$, where $\sigma_k^2$ is the variance of matrix $A$ when initialized with Kaiming initialization. `Init[AB]`-`Init[A]` denotes the accuracy improvement achieved by non-zero initlization, where $B$ is initialized with the same variance as $A$.

there is no need to subtract the non-zero initialized LoRA from the pretrained model. Further verification will be conducted in subsequent experiments.

## 4. Experiments

In this section, we evaluate our findings across three benchmarks. First, we evaluate natural language understanding tasks using a subset of the GLUE benchmark (Wang et al., 2019). Next, we evaluate natural language generation tasks using the commonsense reasoning and arithmetic reasoning benchmarks, as proposed in (Hu et al., 2023). Our primary objective is to examine the impact of zero and non-zero initialization on fine-tuning performance across different learning rates and initialization variances. Let $\sigma_k^2$ represent the variance of matrix $A$ under Kaiming initialization (i.e., $\sigma_k^2 = \frac{1}{n}$). In our experiments, the initialization variance is set to $(\beta\sigma_k)^2$, where $\beta$ controls the initialization size. For non-zero initialization, $B$ is initialized with the same variance as $A$. By default, we subtract the non-zero initialized LoRA from the pretrained weights. The effects of omitting this subtraction process will be discussed in Section 4.3. More experimental details and results can be found in Appendices D and E. It is worth noting that the standard deviation for the GLUE dataset ranges from 0.01 to 0.4, while for commonsense and arithmetic reasoning tasks, it spans from 0.2 to 0.4. In our experiments, smaller learning rates tend to converge less effectively and exhibit higher standard deviations. However, this standard deviation

is generally minimal compared to the performance gains achieved by non-zero initialization.

### 4.1. Natural Language Understanding Tasks

**Models and Datasets.** In this section, we fine-tune the T5-Base model (Raffel et al., 2020) on several datasets from the GLUE benchmark, including MRPC, CoLA, SST-2, and QNLI. The experiment is conducted based on the code provided in (Wang et al., 2024). Each experiment is run three times, and the average test accuracy is reported.

**Results.** Figure 4 presents the test accuracy of T5-Base on the GLUE benchmark. As the initialization variance increases, `Init[A]` exhibits enhanced robustness at smaller learning rates, aligning with our theoretical predictions. Notably, $\beta = 1$ corresponds to the Kaiming initialization ($\gamma[A_0] = -1$), which is smaller than the theoretically optimal value of $\gamma[A_0] = -\frac{1}{2}$. Moreover, initializing $B$ with the same variance as $A$ (i.e., `Init[AB]`) further improves performance over `Init[A]` at smaller learning rates, resulting in up to a 10% accuracy improvement. Figure 5 illustrates the convergence curves of `Init[A]` and `Init[AB]` on the QNLI dataset with a learning rate of 3e-7, where `Init[AB]` accelerates convergence by nearly a factor of two. Additionally, even when the learning rate is near optimal, `Init[AB]` still yields a 1% accuracy improvement. A heatmap for `Init[AB]` is provided in Appendix E.

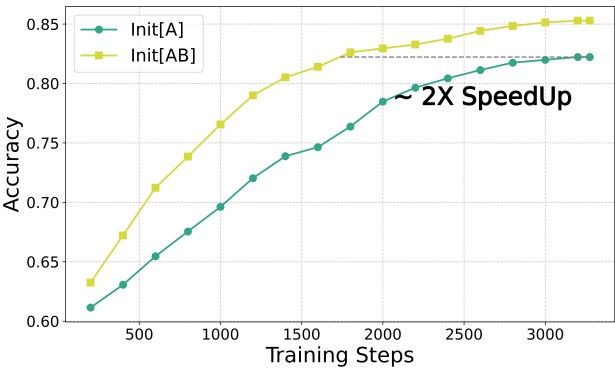

*Figure 5.* Test accuracy of T5-Base finetuned on the QNLI dataset with learning rate $\eta = 3e - 7$ and the initlization size $\beta = 16$.

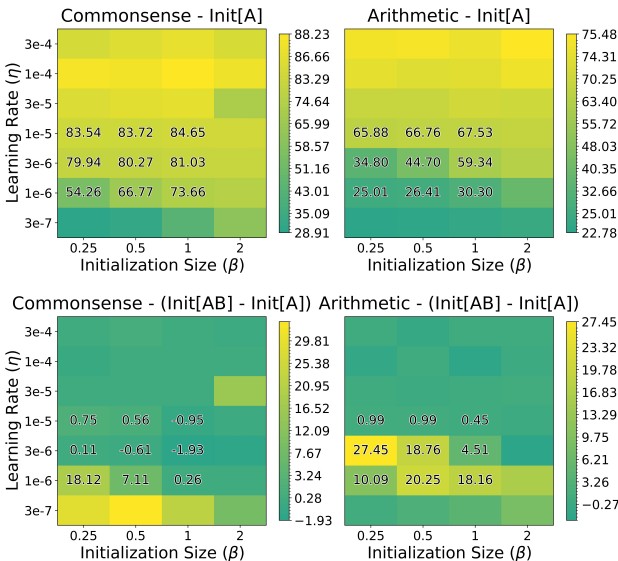

*Figure 6.* Fine-tuning accuracy of Llama 3-8B.

## 4.2. Natural Language Generation Tasks.

**Models and Datasets.** In this section, we fine-tune the Llama 3-8B model (Dubey et al., 2024) on commonsense and arithmetic reasoning benchmarks. The former contains eight subtasks, while the latter contains six subtasks. Each subtask is associated with a predefined training and test set. The experiment is conducted using the code provided in (Hu et al., 2023). Following the procedure outlined in (Hu et al., 2023), we combine the training datasets of all subtasks to create a final training dataset and evaluate each task using a single test dataset. The average test accuracy is reported, and task-specific accuracies are provided in Appendix E.

**Results.** Figure 6 illustrates the average test accuracy of Llama 3-8B on commonsense and arithmetic reasoning benchmarks. The results are consistent with those observed on the GLUE benchmark, where `Init[AB]` generally outperforms `Init[A]`, particularly at smaller learning rates. Moreover, the performance improvement observed with `Init[AB]` on these two benchmarks is notably greater than that on the GLUE dataset. This is likely due to the increased complexity of these two benchmarks, which impose higher demands on the learning rate robustness.

## 4.3. Ablation on the Starting Point of Fine-Tuning

By default, we subtract the non-zero initialized $AB$ from the pretrained weights $W$ to ensure that fine-tuning starts from the pretrained model. In this section, we investigate whether this subtraction is necessary. The strategy of non-zero initialization without this subtraction process is denoted as `Init[AB+]`. The advantage of `Init[AB+]` is that it avoids modifying the pretrained model, rather than improving accuracy. Figure 7 illustrates the differences between `Init[AB+]` and `Init[AB]` under varying initialization variances. When the initialization variance exceeds a certain threshold ($\beta \geq 8$), `Init[AB+]` leads to a significant accu-

racy drop due to excessive noise. However, when suitable initialization variances are applied, there is no discernible difference between `Init[AB+]` and `Init[AB]`. Furthermore, Figure 7 demonstrates that the range of suitable initialization variances is relatively wide, encompassing the variance used in the Kaiming initialization. More ablation results for different models and datasets are in Appendix E.

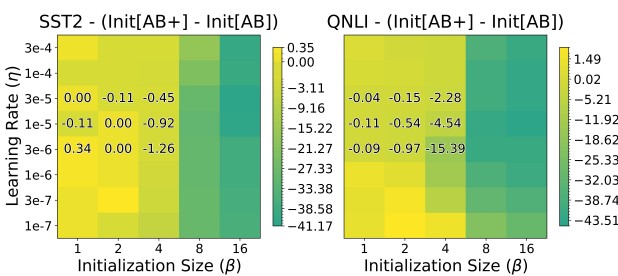

*Figure 7.* Fine-tuning accuracy with different starting points. `Init[AB+]` denotes the variant which do not subtract non-zero initialized $AB$ from $W$, and `Init[AB+]`-`Init[AB]` indicates the accuracy gap between these two initialization strategies.

## 5. Related Works

### 5.1. Low-Rank Adaptation

The fine-tuning performance of LoRA is influenced by several factors, such as rank, gradient, learning rate, and initialization. AdaLoRA (Zhang et al., 2023) adaptively adjusts the rank across different LoRA layers to improve efficiency. Zeng & Lee (2024) provide theoretical guidelines for select-

ing an appropriate rank in LoRA. BoRA (Li et al., 2025) adopts a block-wise mechanism in LoRA, leveraging block matrix multiplication with block-wise diagonal matrices to enhance its expressivity. LoRA-Pro (Wang et al., 2025) improves LoRA by adjusting low-rank gradients to better match full fine-tuning, reducing the performance gap. Additionally, LoRA+ (Hayou et al., 2024a) analyzes how a larger learning rate for the matrix $B$ promotes more stable and efficient LoRA fine-tuning. A key advantage of LoRA+ is its ability to achieve internal stability, i.e., $Z_A = \Theta(1)$, which cannot be achieved using a uniform learning rate for $A$ and $B$. Notably, LoRA+ can also be applied to non-zero initialization to ensure internal stability, as further discussed in Appendix B.2.

Hayou et al. (2024b) have analyzed the effect of initialization on LoRA's fine-tuning dynamics, though their analysis is limited to zero initialization. In contrast, we further investigate and demonstrate that non-zero initialization can enhance LoRA's robustness to suboptimal learning rates. In fact, non-zero initialization has several practical applications, such as initializing the matrices $A$ and $B$ with quantization errors (Li et al., 2024), the main components of pretrained weights (Meng et al., 2024), or the gradients of pretrained weights derived from several data samples (Wang et al., 2024). Recent work (Ponkshe et al., 2024) has highlighted the benefits of using approximate gradients for LoRA initialization. LoRA-One (Zhang et al., 2025) leverages spectral initialization to achieve subspace alignment and generalization guarantees at initialization, enabling efficient convergence from the very start.

However, these studies overlook a crucial distinction between their approaches and standard LoRA, namely, the difference between zero and non-zero initialization. Unlike LoRA-GA and PiSSA, which were motivated by intuition, our approach is motivated by the theoretical analysis of LoRA's fine-tuning dynamics. From the solution set in Eqs.(5-6), we observe that stable and efficient learning imposes stricter constraints on learning rates, whereas the initialization space is more flexible. Traditional zero initialization ($\gamma[B_0] = -\infty$) is merely an extreme case. This motivates us to reconsider the necessity of zero initialization and explore the potential benefits of non-zero initialization. Notably, our motivation and claims are not in competition with prior non-zero initialization methods, such as LoRA-GA and PiSSA. Instead, our findings offer a theoretical foundation for their effectiveness and provide an explanation for the observed performance improvements.

**5.2. Other Parameter-Efficient Fine-Tuning Methods**

In addition to LoRA, two widely adopted PEFT strategies are adapter-based and soft prompt-based approaches. Adapter-based methods (Houlsby et al., 2019; He et al.,

2022; Wang et al., 2022) introduce small trainable modules into the pretrained model and update only these components, significantly reducing training costs. However, the added layers can increase inference latency. In contrast, soft prompt-based methods (Lester et al., 2021; Li & Liang, 2021; Razdaibiedina et al., 2023) prepend task-specific, learnable tokens to the input, leveraging the pretrained model's existing knowledge for adaptation. Despite their efficiency, these methods also incur additional computational overhead and inference delays. By comparison, LoRA supports manual integration of weight updates into the pretrained weights post fine-tuning, thereby avoiding extra inference latency. Additionally, sparse fine-tuning methods such as BitFit (Zaken et al., 2022) update only the model's bias terms, offering another lightweight alternative.

## 6. Conclusion

In this paper, we revisit the fine-tuning dynamics of LoRA and identify that stable and efficient fine-tuning requires specific conditions on both the learning rate and initialization. Compared to initialization, LoRA imposes stricter requirements on the learning rate. Further analysis reveals that initializing both $A$ and $B$ of LoRA to non-zero values (i.e., non-zero initialization) can improve its robustness to suboptimal learning rates, particularly small ones. Note that this is a zero-cost adjustment of LoRA. Additionally, we examine the noise introduced by non-zero initialization for pretraining weights. Extensive experiments demonstrate that with appropriate initialization variances, the noise from non-zero initialization can be effectively corrected through fine-tuning, without negatively impacting fine-tuning performance. Our experiments reveal that the range of suitable initialization variances is relatively broad, encompassing commonly used methods like Kaiming initialization.

## Acknowledgements

This work is supported by the National Key Research and Development Program of China under grant 2024YFC3307900; the National Natural Science Foundation of China under grants 62376103, 62302184, 62436003, and 62206102; Major Science and Technology Project of Hubei Province under grant 2024BAA008; Hubei Science and Technology Talent Service Project under grant 2024DJC078; and Ant Group through CCF-Ant Research Fund. The computation is completed in the HPC Platform of Huazhong University of Science and Technology.

## Impact Statement

This paper explores the non-zero initialization of LoRA, which has the potential to improve the performance of LoRA fine-tuning. Our work builds upon LoRA, and we assert that it does not introduce any negative social implications that would necessitate further discussion.

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

# A. Background

## A.1. Scaling Theory of Neural Networks

In neural networks, scaling refers to enhancing the model's representational capacity by increasing the scale of a specific structure within the model (Kaplan et al., 2020; Hoffmann et al., 2022). It involves analyzing and optimizing several factors, such as model depth, width, training sample size, and the number of training steps. This paper focuses on scaling the model width, which holds practical significance, as the most advanced LLMs today employ extremely large widths, with hidden layer dimensions often reaching tens of thousands (OpenAI, 2024; DeepSeek-AI, 2024).

As the width $n$ increases, the synergistic adaptation of parameter initialization and training dynamics becomes increasingly critical. Previous studies have demonstrated that the variance of parameters should be scaled by $1/n$ to prevent gradient instability during optimization (He et al., 2015). These scaling rules are typically derived from asymptotic analyses of key quantities (e.g., pre-activation values), which theoretically guide adjustments to parameter initialization schemes, learning rates, and even network architectures (Hayou et al., 2019; Schoenholz et al., 2017; Yang, 2019).

Building on this foundation, Yang et al. (2022) proposed the Maximal Update Parameterization ($\mu P$), which constructs scaling rules for initialization schemes, learning rates, and network architectures to theoretically guarantee stability and maximal feature learning in the limit of infinite network width. The key conditions for $\mu P$ are as follows: 1) Stability: for all layers $l$, $Y_l = \Theta(1)$, where the asymptotic notation $\Theta(\cdot)$ is defined in Section 2.1; 2) Feature learning: $\Delta Y_l = \Theta(1)$, where $\Delta$ represents the parameter update after one gradient descent step. In general, $\mu P$ specifies the parameter weights to be randomly initialized at a scale of $\Theta(n^{-1/2})$, with weight updates at a scale of $\Theta(n^{-1})$. The random initialization and update scales for input and output weights should be $\Theta(1)$ and $\Theta(n^{-1})$, respectively. This scaling rule successfully resolves the instability issues inherent in traditional parameterization schemes, making it a key tool in the theoretical analysis and practical application of large models.

## A.2. The $\gamma$-operator

In neural network scaling theory, the asymptotic behavior of key quantities is typically analyzed when scaling specific model components. For example, when scaling the network width, the objective is to quantify how certain network properties evolve as the width $n$ grows. In this context, asymptotic notations, such as $\Theta(\cdot)$, are particularly useful, as defined in Section 2.1. This standard approach has been employed to derive scaling rules for various components, such as initialization (Schoenholz et al., 2017), activation functions (Hayou et al., 2019), and network parameterization (Yang et al., 2024).

Assuming that the weights are initialized as $\Theta(n^{-\beta})$ ($\beta \geq 0$), and that the learning rate scales polynomially with respect to the width $n$, it follows that key quantities, such as pre-activations, gradients, and weight updates, are also asymptotically polynomial in $n$. This regularity arises from the aggregate nature of neural network operations, where tensor operations involve linear combinations of numerous parameters. Consequently, we introduce the $\gamma$-operator to express the polynomial behavior as $v = \Theta(\gamma[v])$. We now define some basic operations using the $\gamma$-operator.

**Zero**: $\gamma[0] = -\infty$, corresponding to the limit of $\gamma[n^\beta]$ when $\beta \to -\infty$.

**Product**: For two real-valued variables $v, v' \in \mathbb{R}$, we have $\gamma[vv'] = \gamma[v] + \gamma[v']$.

**Addition**: For two real-valued variables $v, v' \in \mathbb{R}$ and $v \neq -v'$, we have $\gamma[v + v'] = \max(\gamma[v], \gamma[v'])$.

# B. Fine-Tuning Dynamics of LoRA with Adam

## B.1. Detailed Proof

In this section, we present a detailed explanation of the process from Eq. (3) to Eq. (5). To facilitate the analysis, we first introduce Lemmas 1 and 2.

**Lemma 1.** *Let $U \in \mathbb{R}^{1 \times r}$ and $V \in \mathbb{R}^r$, where $r$ is independent of $n$. If $U = \Theta(n^\alpha)$ and $V = \Theta(n^\beta)$, then $UV = \Theta(n^{\alpha+\beta})$.*

**Proof of Lemma 1.** By the definition of the notation $\Theta$, each element of $U$ and $V$ satisfies the following relations:

$$U_i = \Theta(n^\alpha), \quad V_i = \Theta(n^\beta), \quad i \in [r] \tag{7}$$

There exist positive constants $\kappa_u^l, \kappa_u^h, \kappa_v^l$ and $\kappa_v^h$ such that

$$\kappa_u^l n^\alpha < U_i < \kappa_u^h n^\alpha, \quad \kappa_v^l n^\beta < V_i < \kappa_v^h n^\beta, \tag{8}$$

Multiplying $U_i$ and $V_i$, we obtain

$$\kappa_u^l \kappa_v^l n^{\alpha+\beta} < U_i V_i < \kappa_u^h \kappa_v^h n^{\alpha+\beta} \tag{9}$$

Thus, the product $UV$ satisfies the following inequality:

$$r\kappa_u^l \kappa_v^l n^{\alpha+\beta} < UV = \sum_{i=1}^{r} U_i V_i < r\kappa_u^h \kappa_v^h n^{\alpha+\beta} \tag{10}$$

Therefore, by the definition of the notation $\Theta$, we conclude that

$$UV = \Theta(\alpha+\beta), \quad \gamma[UV] = \alpha + \beta \tag{11}$$

Note that the same reasoning and result hold for the matrix multiplication of $U \in \mathbb{R}^{a \times r}$ and $U \in \mathbb{R}^{r \times b}$.

**Lemma 2.** *Let $U \in \mathbb{R}^{1 \times n}$ and $V \in \mathbb{R}^n$. If $U = \Theta(n^\alpha)$ and $V = \Theta(n^\beta)$, then $UV = \Theta(n^{\alpha+\beta+1})$.*

**Proof of Lemma 2.** Similar to the proof of Lemma 1, the product $UV$ satisfies the following inequality:

$$\kappa_u^l \kappa_v^l n^{\alpha+\beta+1} < UV = \sum_{i=1}^{n} U_i V_i < \kappa_u^h \kappa_v^h n^{\alpha+\beta+1} \tag{12}$$

Thus, by the definition of the notation $\Theta$, we conclude that

$$UV = \Theta(\alpha+\beta+1), \quad \gamma[UV] = \alpha + \beta + 1 \tag{13}$$

Note that the same reasoning and result hold for the matrix multiplication of $U \in \mathbb{R}^{a \times n}$ and $U \in \mathbb{R}^{n \times b}$.

**Proof of the process from Eq.(3) and Eq.(5).** Eq. (3) outlines the conditions for LoRA to achieve stability and efficiency, as follows:

$$\begin{cases} \gamma[\delta_t^1] = \gamma[-\eta_A B_{t-1} g_A^{t-1} \underline{Z}] = 0 & (\delta_t^1 = \Theta(1)) \\ \gamma[\delta_t^2] = \gamma[-\eta_B g_B^{t-1} A_{t-1} \underline{Z}] = 0 & (\delta_t^2 = \Theta(1)) \\ \gamma[Z_B^{t-1}] = \gamma[B_{t-1} A_{t-1} \underline{Z}] = 0 & (Z_B^{t-1} = \Theta(1)) \end{cases} \tag{14}$$

Taking $\delta_t^1$ as an example, where $\eta_A \in \mathbb{R}$, $B_{t-1} \in \mathbb{R}^{n \times r}$, $g_A \in \mathbb{R}^{r \times n}$, and $\underline{Z} \in \mathbb{R}^n$, we expand $\gamma[\delta_t^1]$ as follows:

$$\begin{aligned} \gamma[\delta_t^1] &= \gamma[-\eta_A B_{t-1} g_A^{t-1} \underline{Z}] \\ &\overset{(a)}{\leq} \gamma[\eta_A] + \gamma[B_{t-1} g_A^{t-1} \underline{Z}] \\ &\overset{(b)}{\leq} \gamma[\eta_A] + \gamma[B_{t-1}] + \gamma[g_A^{t-1} \underline{Z}] \\ &\overset{(c)}{\leq} \gamma[\eta_A] + \gamma[B_{t-1}] + \gamma[g_A^{t-1}] + \gamma[\underline{Z}] + 1 \\ &= \gamma[\eta_A] + \gamma[B_0 + \eta_B \sum_{i=0}^{t-2} g_B^i] + \gamma[g_A^{t-1}] + \gamma[\underline{Z}] + 1 \\ &\overset{(d)}{\leq} \gamma[\eta_A] + \max(\gamma[B_0], \gamma[\eta_B]) + 1, \end{aligned} \tag{15}$$

where (a) follows from the multiplication properties of the $\gamma$-operator, (b) follows from Lemma 1, (c) follows from Lemma 2, and (d) follows from the fact that $\gamma[g_A] = \gamma[g_B] = \gamma[\underline{Z}] = 0$.

Similarly, performing the same analysis on $\delta_t^2$ and $Z_B$ yields the following relationships:

$$\begin{cases} \gamma[\delta_t^1] = \max(\gamma[B_0], \gamma[\eta_B]) + \gamma[\eta_A] + 1 = 0, \\ \gamma[\delta_t^2] = \max(\gamma[A_0], \gamma[\eta_A]) + \gamma[\eta_B] + 1 = 0, \\ \gamma[Z_B^{t-1}] = \max(\gamma[A_0], \gamma[\eta_A]) + \max(\gamma[B_0], \gamma[\eta_B]) + 1 = 0. \end{cases} \tag{16}$$

Furthermore, we obtain:

$$\gamma[\delta_t^1] + \gamma[\delta_t^2] - \gamma[Z_B^{t-1}] = \gamma[\eta_A] + \gamma[\eta_B] + 1 = 0. \tag{17}$$

Substituting this equation back into $\gamma[\delta_t^1]$ and $\gamma[\delta_t^2]$ yields:

$$\begin{cases} \max(\gamma[A_0], \gamma[\eta_A]) = \gamma[\eta_A], \\ \max(\gamma[B_0], \gamma[\eta_B]) = \gamma[\eta_B]. \end{cases} \tag{18}$$

Thus, we conclude that:

$$\begin{cases} \gamma[\eta_A] + \gamma[\eta_B] = -1, \\ \gamma[A_0] \leq \gamma[\eta_A], \quad \gamma[B_0] \leq \gamma[\eta_B]. \end{cases} \tag{19}$$

### B.2. Internal Stability

In this paper, we focus solely on the stability of LoRA's final output, $Z_B$, while excluding the stability of LoRA's internal output, $Z_A$. Hu et al. (2022) discusses $Z_A$ and demonstrates that init[B] ensures stability for both $Z_A$ and $Z_B$, but fails to facilitate efficient updates of $Z_B$, as defined in Definition 4. In contrast, init[A] achieves stability and efficient updating of $Z_B$, but does not ensure the stability of $Z_A$. Here, we additionally consider the internal stability by introducing the condition $Z_A^{t-1} = A_{t-1}\underline{Z} = \Theta(1)$, as a requirement in Eq. (3).

$$\begin{cases} \gamma[\delta_t^1] = \gamma[-\eta_A B_{t-1} g_A^{t-1} \underline{Z}] = 0 & (\delta_t^1 = \Theta(1)) \\ \gamma[\delta_t^2] = \gamma[-\eta_B g_B^{t-1} A_{t-1} \underline{Z}] = 0 & (\delta_t^2 = \Theta(1)) \\ \gamma[Z_B^{t-1}] = \gamma[B_{t-1} A_{t-1} \underline{Z}] = 0 & (Z_B^{t-1} = \Theta(1)) \\ \gamma[Z_A^{t-1}] = \gamma[A_{t-1} \underline{Z}] = 0 & (Z_A^{t-1} = \Theta(1)) \end{cases} \tag{20}$$

Similar to the proof in Section B.1, we can express the above equations in terms of $(A_0, B_0)$ and $(\eta_A, \eta_B)$ as follows:

$$\begin{cases} \max(\gamma[B_0], \gamma[\eta_B]) + \gamma[\eta_A] + 1 = 0, \\ \max(\gamma[A_0], \gamma[\eta_A]) + \gamma[\eta_B] + 1 = 0, \\ \max(\gamma[A_0], \gamma[\eta_A]) + \max(\gamma[B_0], \gamma[\eta_B]) + 1 = 0, \\ \max(\gamma[A_0], \gamma[\eta_A]) + 1 = 0. \end{cases}$$

Based on the derivation steps outlined in Appendix B.1, the solution set can be obtained as follows:

$$\begin{cases} \gamma[\eta_A] = -1, \gamma[\eta_B] = 0, \\ \gamma[A_0] \leq \gamma[\eta_A] = -1, \quad \gamma[B_0] \leq \gamma[\eta_B] = 0, \end{cases} \tag{21}$$

where $\gamma[A_0]$ and $\gamma[B_0]$ cannot both be equal to $-\infty$ simultaneously, as this would imply that both $A$ and $B$ are zero, preventing optimization via gradient descent algorithms.

This is actually the solution proposed in LoRA+ (Hayou et al., 2024a), which sets $\gamma[\eta_A] = -1, \gamma[\eta_B] = 0$. Similar to the analysis in Section 3.3, setting $\gamma[A_0] = -1$ and $\gamma[B_0] = 0$ can achieve the Robust LoRA defined in Theorem 1. Thus, non-zero initialization can also improve the robustness of LoRA+ in relation to the learning rate.

## C. Fine-Tuning Dynamics of LoRA with SGD

In this section, we examine the fine-tuning dynamics of LoRA using the SGD optimizer. For simplicity, we consider the model described in LoRA+ (Hayou et al., 2024a), which can be expressed as:

$$f(x) = ba^\top x, \tag{22}$$

where the pretrained weights are assumed to be zero, $x \in \mathbb{R}^n$ is the input vector, and $b \in \mathbb{R}, a \in \mathbb{R}^n$ are the LoRA weights. Additionally, we assume that the fine-tuning dataset consists of a single data point, $(x, y)$, with $x = \Theta(1)$. The objective is to minimize the loss function $\mathcal{L}(x) = \frac{1}{2}(f(x) - y)^2$. The corresponding gradients are as follows:

$$g_a^t = b_t(f_t(x) - y)x, \quad g_b^t = a_t^\top x(f(x) - y). \tag{23}$$

Using Lemmas 1 and 2, we have

$$\gamma[g_a^t] = \gamma[b^t], \quad \gamma[g_b^t] = \gamma[a^t] + 1. \tag{24}$$

Notably, the Adam optimizer guarantees that $g_A = \Theta(1)$ and $g_B = \Theta(1)$ through gradient normalization, whereas SGD does not ensure this. Let $U_t = f_t(x) - y$. At step $t$, the update of LoRA's output is given by:

$$\Delta f_t = f_t(x) - f_{t-1}(x) = \underbrace{- \eta_a b_{t-1}^2 U_{t-1} \|x\|^2}_{\delta_t^1} \underbrace{- \eta_b (a_{t-1}^\top x)^2 U_{t-1}}_{\delta_t^2} + \underbrace{\eta_a \eta_b U_{t-1}^2 b_{t-1} (a_{t-1}^\top x) \|x\|^2}_{\delta_t^3}. \tag{25}$$

To ensure stability and efficiency as defined in Definitions 3 and 4, the following conditions must hold for $t > 1$:

$$\begin{cases} \gamma[\delta_t^1] = \gamma[\eta_a b_{t-1}^2 U_{t-1} \|x\|^2] = \gamma[\eta_a] + 2\gamma[b_{t-1}] + 1 = 0 & (\delta_t^1 = \Theta(1)) \\ \gamma[\delta_t^2] = \gamma[\eta_b (a_{t-1}^\top x)^2 U_{t-1}] = \gamma[\eta_b] + 2\gamma[a_{t-1}^\top] + 2 = 0 & (\delta_t^2 = \Theta(1)) \\ \gamma[f_{t-1}(x)] = \gamma[b_{t-1} a_{t-1}^\top x] = \gamma[b_{t-1}] + \gamma[a_{t-1}] + 1 = 0 & (f_{t-1}(x) = \Theta(1)) \end{cases} \tag{26}$$

Further, we have the following relation:

$$\gamma[\delta_t^1] + \gamma[\delta_t^2] - 2\gamma[f_{t-1}(x)] = \gamma[\eta_a] + \gamma[\eta_b] + 1 = 0, \tag{27}$$

which is the same as the result obtained with the Adam optimizer. Using a uniform learning rate for both $a$ and $b$, we have

$$\begin{cases} \gamma[\eta_a] = \gamma[\eta_b] = -\frac{1}{2}, \\ \gamma[a_{t-1}] = -\frac{3}{4}, \quad \gamma[b_{t-1}] = -\frac{1}{4}, \quad (t > 1). \end{cases} \tag{28}$$

However, the values of $\gamma[a_0]$ and $\gamma[b_0]$ are unknown. Therefore, we take $a_1 = a_0 + \eta_a g_a^0$ and $b_1 = b_0 + \eta_b g_b^0$, leading to

$$\begin{cases} \gamma[a_1] = \max\{\gamma[a_0], \gamma[\eta_a] + \gamma[g_a^0]\} = \max\{\gamma[a_0], -\frac{1}{2} + \gamma[b^0]\} = -\frac{3}{4}, \\ \gamma[b_1] = \max\{\gamma[b_0], \gamma[\eta_b] + \gamma[g_b^0]\} = \max\{\gamma[b_0], -\frac{1}{2} + \gamma[a^0] + 1\} = -\frac{1}{4}. \end{cases} \tag{29}$$

Solving this system of equations yields the following conditions:

$$\begin{cases} \gamma[a_0] \leq -\frac{3}{4}, \\ \gamma[b_0] \leq -\frac{1}{4}, \end{cases} \tag{30}$$

where at least one of the inequalities must hold with equality. Notably, when both inequalities hold with equality, the conditions for robust LoRA, as defined in Theorem 1, are satisfied. Specifically, when $\gamma[a_0] = -\frac{3}{4}$, and $\gamma[b_0] = -\frac{1}{4}$, for all $\gamma[\eta_a] < -1/2, \gamma[\eta_b] < -1/2$, we have the following result:

$$\begin{cases} \gamma[\delta_t^1] = \gamma[\eta_a] + \frac{1}{2}, \\ \gamma[\delta_t^2] = \gamma[\eta_b] + \frac{1}{2}, \\ \gamma[f_{t-1}(x)] = 0, \end{cases} \tag{31}$$

where $\gamma[\delta_t^1]$ depends only on $\gamma[\eta_a]$, $\gamma[\delta_t^2]$ depends only on $\gamma[\eta_b]$, and $\gamma[f_{t-1}(x)]$ is independent of both $\gamma[\eta_a]$ and $\gamma[\eta_b]$.

## D. Experimental Settings

### D.1. Toy Model

- **Model and Dataset**: The toy model used in Section 3.4 is defined as follows:

$$f(X) = W_{out} \mathcal{F}((W_0 + BA)\mathcal{F}(W_{in}X)),$$

where $X \in \mathbb{R}^d$ is the input data, $W_{in} \in \mathbb{R}^{n \times d}$, $W_0 \in \mathbb{R}^{n \times n}$, and $W_{out} \in \mathbb{R}^{10 \times n}$ are the pretrained weights, $A \in \mathbb{R}^{r \times n}$ and $B \in \mathbb{R}^{n \times r}$ are the LoRA weights, and $\mathcal{F}$ denotes the ReLU activation function(Glorot et al., 2011). The MNIST (Deng, 2012) and FashionMNIST (Xiao et al., 2017) datasets are used for pretraining and fine-tuning, with input images flattened into one-dimensional vectors. The model dimensions $(d, n, r)$ are set to $(784, 4096, 32)$.

- **Training Algorithm**: Both pretraining and fine-tuning employ the AdamW optimizer (Loshchilov & Hutter, 2019), with $\beta_1 = 0.9$, $\beta_2 = 0.999$, $\epsilon = 1e - 8$, and a weight decay of 0. The batch size is set to 64. During pretraining, the learning rate is set to $1e - 3$, with 2000 training steps. During fine-tuning, the learning rate is tuned among the set $\{3e - 4, 4e - 4, 5e - 4, 6e - 4, 7e - 4, 8e - 4, 9e - 4, 1e - 3, 2e - 3, 3e - 3\}$, with $W_0, W_{in}, W_{out}$ frozen. Each learning rate is trained for 100 steps. All experiments are conducted with 10 independent repetitions using different random seeds, and the final results are averaged over the test accuracy.

- **Initialization Strategy**: During pretraining, $W_0, W_{in}, W_{out}$ are initialized using Kaiming initialization (He et al., 2015). During fine-tuning, $A$ and $B$ are initialized with a Gaussian distribution, with mean 0 and standard deviations $\sigma_A$ and $\sigma_B$, respectively. For `Init[A]`, $\sigma_B$ is set to 0, and $\sigma_A$ is tuned among the set $\{2e - 5, 5e - 5, 8e - 5, 1e - 4, 4e - 4, 7e - 4, 1e - 3, 3e - 3, 6e - 3, 9e - 3, 2e - 2, 5e - 2\}$. For `Init[AB]`, $\sigma_A$ and $\sigma_B$ are kept equal, and both standard deviations are tuned among the same set. Note that the non-zero initialized $AB$ is not aubstracted from $W_0$.

### D.2. Experiments on Natural Language Understanding.

- **Model and Dataset**: T5-Base (Raffel et al., 2020), with sequence length $T = 128$, is fine-tuned on several datasets from the GLUE benchmark (Wang et al., 2019), including MRPC, CoLA, SST-2, and QNLI.

- **LoRA Hyperparameters**: The rank is set to $r = 16$ and $\alpha = 16$. LoRA is applied to the 'query' and 'value' weights, using full precision (FP32).

- **Training Algorithm**: AdamW (Loshchilov & Hutter, 2019) with $\beta_1 = 0.9, \beta_2 = 0.999, \epsilon = 1e - 8$ and weight decay of 0. The learning rate is tuned among the set $\{1e - 7, 3e - 7, 1e - 6, 3e - 6, 1e - 5, 3e - 5, 1e - 4, 3e - 4\}$, a warmup ratio of 0.03 is employed. Number of train epochs $E = 1$.

- **Initialization Strategy**: The initialization variance is set to $(\beta\sigma_k)^2$, where $\sigma_k^2$ represent the variance of matrix $A$ under Kaiming initialization (i.e., $\sigma_k^2 = \frac{1}{n}$). $\beta$ is tuned among $\{1, 2, 4, 8, 16\}$. For non-zero initialization, $B$ is initialized with the same variance as $A$. By default, we subtract the non-zero initialized LoRA from the pretrained weights.

### D.3. Experiments on Natural Language Generation.

- **Model and Dataset**: Llama 3-8B (Dubey et al., 2024), with sequence length $T = 256$, is fine-tuned on the common-sense reasoning and arithmetic reasoning benchmarks, as proposed in (Hu et al., 2023).

- **LoRA Hyperparameters**: The rank is set to $r = 16$ and $\alpha = 16$. LoRA is applied to the 'q_proj', 'k_proj', 'v_proj', 'up_proj', and 'down_proj' weights, using full precision (FP32).

- **Training Algorithm**: AdamW (Loshchilov & Hutter, 2019) with $\beta_1 = 0.9, \beta_2 = 0.999, \epsilon = 1e - 8$ and weight decay of 0. The learning rate is tuned among the set $\{3e - 7, 1e - 6, 3e - 6, 1e - 5, 3e - 5, 1e - 4, 3e - 4\}$, a warmup ratio of 0.03 is employed, and linear decay are employed. Number of train epochs $E = 1$.

- **Initialization Strategy**: The initialization variance is set to $(\beta\sigma_k)^2$, where $\sigma_k^2$ represent the variance of matrix $A$ under Kaiming initialization. $\beta$ is tuned among $\{0.25, 0.5, 1, 2, 4\}$. For non-zero initialization, $B$ is initialized with the same variance as $A$. By default, we subtract the non-zero initialized LoRA from the pretrained weights.

## E. Additional Experimental Results

In Section 4, we present the accuracy heatmaps for `Init[A]` and `Init[AB]-Init[A]`. Here, we directly display the heatmaps for `Init[AB]`, including for the T5-Base model, as shown in Figure 8, and for the Llama 3-8B model, as depicted in Figure 9. Additionally, we provide the accuracy gap between `Init[AB+]` and `Init[AB]` on the MRPC and CoLA datasets. As shown in Figure 10, there is no significant difference between `Init[AB+]` and `Init[AB]` under appropriate initialization variance. To verify the impact of non-zero initialization on LoRA+ (Hayou et al., 2024a), we compared the accuracy of `Init[AB]` and `Init[A]` under various settings. Figure 11 that non-zero initialization can also improve LoRA+'s robustness and accuracy.

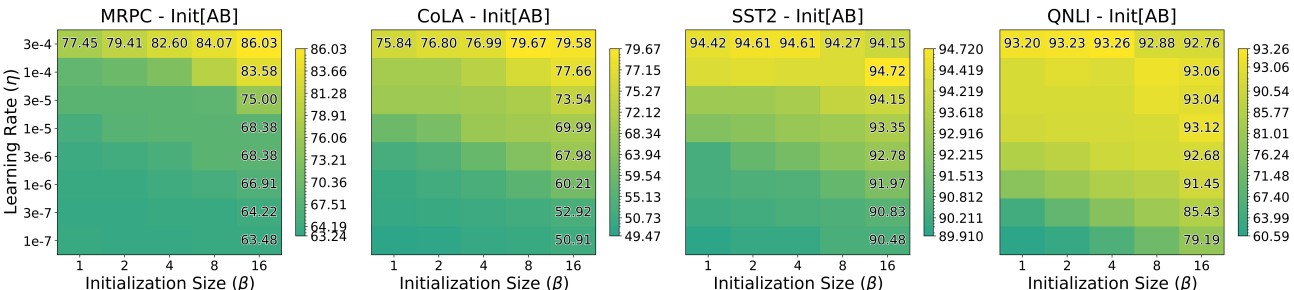

*Figure 8.* Fine-tuning accuracy of T5-Base on the GLUE benchmark with non-zero initialization.

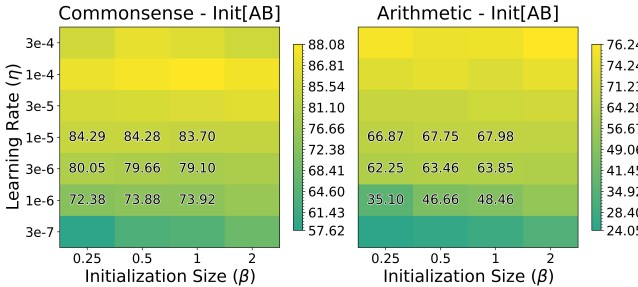

*Figure 9.* Fine-tuning accuracy of Llama 3-8B on the commonsense and arithmetic reasoning benchmarks with non-zero initialization.

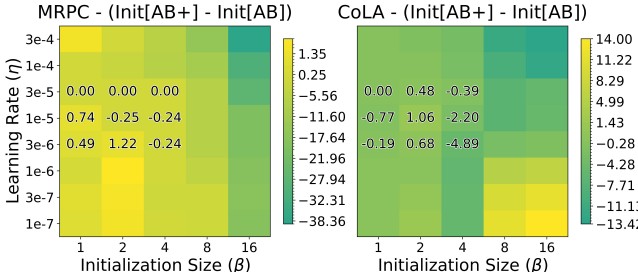

*Figure 10.* Fine-tuning accuracy with different starting points. `Init[AB+]` denotes the variant which do not subtract non-zero initialized $A_0 B_0$ from W, and `Init[AB+]-Init[AB]` indicates the accuracy gap between these two initialization strategies.

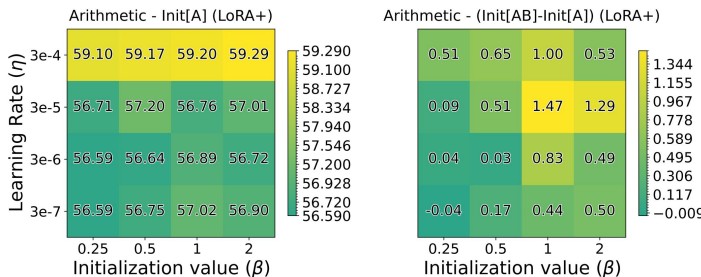

*Figure 11.* Experimental results with LoRA+. The hyperparameter $\lambda$ represents the value of $\eta_B/\eta_A$ (i.e., $\eta_A = \eta, \eta_B = \lambda\eta$). Both `Init[A]` and `Init[AB]` use the initialization size of $\beta = 1$, that is, $\sigma_A^2 = \sigma_B^2 = 1/n$, where $n$ is the network width.

