# OpenReview forum: "Beyond Zero Initialization: Investigating the Impact of Non-Zero Initialization on LoRA Fine-Tuning Dynamics"
_ICML.cc/2025/Conference — ICML 2025 poster_

### Official Review · Reviewer_wxon · 2025-02-19

**Overall Recommendation:** 3

**Summary:**

This paper studies how non-zero initialization improves the perforamnce of LoRA, especially the stabilitiy.

- The authors define 1) the notation of stabilitity, $BAX = \Theta(1)$ for all LoRA layers when the width is infinity, where $X$ is the input. 2) the notation of efficiency, the linear update term is $\Theta(1)$.

- Based on the above two creteria, the author derive the requirements on the random Gaussian variance of A and B as well as the step-size. When using SGD, the optimal initialization under such two creteria is neither classical LoRA initialization nor other variants of zero initialization.

- The author continue to define the robustness of LoRA and derive the similar requirements on them.

**Claims And Evidence:**

This paper looks good and provides some findings beyond zero initialization. I understand the motivation of using non-zero initialization but the current claim is weak. Previous non-zero initialization work, e.g., LoRA-GA, has better motivation. For instance, LoRA-GA is to ensure that LoRA gradient updates can match full gradient updates as much as possible, which is also the spirit of using LoRA.

Under theory-guided instructions, this paper gives some non-zero initialization strategies that achieves better performance than LoRA, e.g., 2x speedup than LoRA. However, the comparison with previous work is limited: only LoRA is compared. I understand that the key idea of this work is to speedup and obtain other benefits (e.g., LoRA). Nevertheless, the experimental comparison is not sufficient.

Another significant issue is that, the derivation heavily follows with previous work, e.g., Hayou et al. When I read it at first, it can be a good journal extension but I'm not sure that it can be regarded as an independent work.

**Essential References Not Discussed:**

The essential references are sufficient but it's true that not main LoRA-based algorithms are discussed.

**Experimental Designs Or Analyses:**

The experiments are not sufficient. Only LoRA is compared.

**Methods And Evaluation Criteria:**

The evaluation makes sense but the comparison is limited.

**Other Comments Or Suggestions:**

N/A

**Other Strengths And Weaknesses:**

N/A

**Questions For Authors:**

N/A

**Relation To Broader Scientific Literature:**

this topic and the obtained findings are interesting to the machine learning community.

**Theoretical Claims:**

The theoretical claim is ok in terms of the stability and efficiency.

---

> ### Author Rebuttal · Authors · 2025-03-31
>
> **Hi Reviewer wxon:**
>
> Thank you for your detailed and insightful comments. Below, we provide responses to each point individually. Additional experimental results can be found in **https://anonymous.4open.science/r/nzlora_rebuttal-7D3E**. To save space, we denote zero initialization as ***ZI*** and non-zero initialization as ***NZI***.
>
> ***Q1: "the current claim about motivation is weak"***
>
> **R1:**
> **Unlike LoRA-GA, which was motivated by intuition, our approach is motivated by the theoretical analysis of LoRA's fine-tuning dynamics.**
> From the solution set in Eqs.(5-6), we observe that stable and efficient learning imposes stricter constraints on learning rates, whereas the initialization space is more flexible. Traditional *ZI* ($\gamma[B_0]=-\infty$) is merely an extreme case. This motivates us to reconsider the necessity of *ZI* and explore the potential benefits of *NZI*. Based on this insight, we conduct further analysis and evaluation, leading to two key findings:
> 1. *NZI* can reduce the sensitivity of LoRA to suboptimal (i.e., smaller) learning rates.
> 2. The purpose of traditional *ZI*, "fine-tuning from a pre-trained model", is not strictly necessary.
>
> Notably, our motivation and claims are not in competition with prior *NZI* methods, such as LoRA-GA and PiSSA. Instead, our findings offer a theoretical foundation for their effectiveness and provide an explanation for the observed performance improvements. Fig.11 in the above link show that the accuracy gains of LoRA-GA and PiSSA primarily stem from *NZI*. These points will be clarified in the revised version of the paper.
>
>
> ***Q2: "the experimental comparison is not sufficient"***
>
> **R2:**
> Following your suggestion, we have added additional comparisons and combinations of LoRA-based methods with *NZI*. Specifically, two key aspects are considered:
>
> 1. **Ablation comparison with LoRA-GA and PiSSA.**
> As shown in Fig.11, a large portion of the accuracy gains in PiSSA and LoRA-GA can be attributed to the use of *NZI*. The remaining gains are due to the fact that initialization values derived from pre-trained weights or gradients are more effective than random noise.
>
> 1. **Combination with LoRA+  and HydraLoRA.** We introduced *NZI* for both LoRA+ (using a larger learning rate for the matrix $B$) and HydraLoRA [1] (an asymmetric LoRA that uses one matrix $A$ and multiple matrices $B$). As shown in Figs.12-13, *NZI* enhances the robustness of LoRA+ and HydraLoRA to variations in learning rate and improves accuracy. The relevant settings are detailed in the figure caption.
>
> [1] HydraLoRA: An Asymmetric LoRA Architecture for Efficient Fine-Tuning, NeurIPS 2024.
>
> ***Q3: "the derivation heavily follows with previous work"***
>
> **R3:**
> The derivation used in this paper, including the notation of $\gamma$ and $\Theta$ and the definitions of stability and efficiency, is widely employed in infinite-width analysis and can be traced back to Yang et al. (NeurIPS 2021; see line 520 of our paper). Hayou et al. used these tools to explore the effects of learning rate (ICML 2024; line 475 in our paper) and initialization (NeurIPS 2024; line 479 in our paper) on zero-initialized LoRA.
> However, a fundamental question remains unaddressed: Why is *ZI* necessary? In this paper, we extend these general derivations to examine the potential advantages of *NZI*.
>
> Notably, **the contribution and innovation of this paper lie not in proposing a new derivation method, but in the following three aspects**:
> 1. **Motivation for *NZI*.** We provided a comprehensive solution set that ensures LoRA's stable and efficient learning. Building on this, we observe that the solution set includes both zero and *NZI*, prompting us to investigate the role of *NZI* further. This constitutes a key distinction between our work and previous studies, where the potential of pure *NZI* has often been overlooked. Our research bridges this gap and provides preliminary evidence supporting the feasibility of *NZI*.
> 2. **A new metric for LoRA's fine-tuning dynamics, "robustness", is proposed.** We compare the fine-tuning dynamics of zero and *NZI*s and define robustness in terms of the sensitivity of these dynamics to the learning rate. The central argument of this paper is that *NZI* exhibits superior robustness compared to *ZI*. We believe that this metric is crucial for LoRA fine-tuning dynamics, offering a significant extension and enhancement to existing theoretical derivations and analyses.
> 3. **Breaking inherent cognitions.** Our analysis and experiments further show that fine-tuning does not need to strictly start from a pre-trained model. This challenges the default practice in previous studies, such as LoRA, LoRA-GA, and PiSSA.
>
> These contributions provide valuable guidance for understanding LoRA initialization and fine-tuning LLMs. They represent notable advancements and are substantial enough to be considered as independent work.

---

> > ### Comment · Reviewer_wxon · 2025-04-05
> >
> > thank for the authors' response with additional experiments.
> >
> > The current explanation looks good for me on the motivation. I sugges the authors to mention it (maybe in a high level way) in the introduction.
> >
> > I understand the authors' claim on "lie not in proposing a new derivation method, but in the following three aspects". NZI has been studied in LoRA-GA, LoRA-Pro as well as (Ponkshe et al., 2024) with some experimental-driven design. This paper claims some theoretical understanding/analysis of NZI. There is one paper posted on arXiv (https://arxiv.org/abs/2502.01235) after ICML deadline which builds a mathematical analysis framework of LoRA under NZI. I suggest the authors to discuss this work in the updated version.
> >
> > Based on the above, I increase my score to 3.

---

> > > ### Author Response · Authors · 2025-04-05
> > >
> > > Thank you again for reviewing our paper and for your valuable feedback! We're glad that the additional experiments and motivation clarification resolved your concerns. Your comments were extremely helpful, and we truly appreciate you increasing the score based on our rebuttal response.
> > >
> > > As suggested, we will enhance the introduction (Section 1) with a more detailed discussion of our motivation to improve clarity for readers.
> > >
> > > Additionally, we sincerely appreciate your suggestion regarding the latest advances in LoRA initialization, particularly the LoRA-One paper on arXiv. We will carefully study these works and incorporate a discussion in our revision to better contextualize our theoretical contributions in relation to these recent developments.

---

### Official Review · Reviewer_rPo6 · 2025-03-10

**Overall Recommendation:** 4

**Summary:**

This paper investigates the impact of non-zero initialization on the fine-tuning dynamics of LoRA. Traditionally, in LoRA, one of the low-rank matrices, A or B, is initialized to zero to ensure fine-tuning starts from the pretrained model. However, this practice lacks theoretical justification. The authors theoretically analyze the effects of initializing both A and B to non-zero values. Their key findings are: (1) Non-zero initialization improves robustness to suboptimal learning rates; and (2) Fine-tuning does not need to strictly start from the pretrained model. The authors validate these findings through extensive experiments across various models and datasets. These results challenge the conventional practice of zero initialization in LoRA and highlight the benefits of non-zero initialization.

**Claims And Evidence:**

The claims made in the paper are theoretically proven and experimentally verified.

**Essential References Not Discussed:**

To the best of the reviewer's knowledge, all related work on LoRA initialization has been covered.

**Experimental Designs Or Analyses:**

The reviewer checked the experimental setup, results, and analysis. As described in the paper, the authors conducted experiments on three standard benchmarks. The experimental setups were based on published work and aligned with general practices in LoRA fine-tuning. The authors primarily analyzed different initialization settings and learning rates, which is consistent with the paper's motivation. The experimental analysis is also reasonable and supports the theoretical findings.

**Methods And Evaluation Criteria:**

The proposed methods and evaluation criteria are reasonable. This paper studies the initialization problem of LoRA, a common but previously overlooked aspect of fine-tuning LLMs. The authors systematically compare different initialization methods (zero vs. non-zero) using models, datasets, and code based on published work, ensuring reliability.

**Other Comments Or Suggestions:**

Typo in line 62: "raise" should be "raises."
Incorrect reference to Llama 3 in line 799.
Typo in Eq (19):  $\gamma[A_0]\leq\eta_A$ should be $\gamma[A_0]\leq\gamma[\eta_A]$.

**Other Strengths And Weaknesses:**

Strengths:
This paper challenges the traditional LoRA initialization method, and studies the significance of non-zero initialization from the perspective of robustness to learning rate. The method is simple yet insightful.

It also fundamentally overturns the purpose of traditional zero initialization (fine-tuning from pre-trained models). Experiments show that non-zero initialization with appropriate variance does not affect fine-tuning performance, indicating that fine-tuning does not need to start strictly from a pretrained model.

Weaknesses:
A minor shortcoming is the lack of discussion on how the definitions of stability and efficiency in this paper differ from those in previous studies (e.g., LoRA+). The authors are encouraged to clarify this distinction in the appendix.

**Questions For Authors:**

How  do the definitions of stability and efficiency in this paper differ from those in previous studies (e.g., LoRA+)?

**Relation To Broader Scientific Literature:**

Previous work (Hayou et al., 2024b) discussed the difference between initializing A or B to zero but did not explore the rationale behind zero initialization. This paper fills that gap, demonstrating that both A and B can be initialized to non-zero values. These findings provide theoretical support for related LoRA variants, such as PiSSA and LoRAGA, and significantly contribute to LoRA research.

**Theoretical Claims:**

The reviewer carefully checked the theoretical claims and corresponding proofs in the paper, including all results in Section 3 and the proofs in Appendices B and C. To the reviewer, the claims and proofs are reasonable. Although there are minor typos, for example, $\gamma[A_0]\leq\eta_A$ in Eq (19) in Appendix B should be $\gamma[A_0]\leq\gamma[\eta_A]$, these do not affect the validity of the theoretical results.

---

> ### Author Rebuttal · Authors · 2025-03-31
>
> **Hi Reviewer rPo6:**
>
> Thank you for your detailed and insightful comments. Below, we provide responses to each point individually. Additional experimental results can be found in **https://anonymous.4open.science/r/nzlora_rebuttal-7D3E**.
>
>
> ***Q1: "typos in lines 62 and 799, and Eq (19)"***
>
> **R1:**
> Thank you for your thorough review. We will correct the identified typos and carefully re-examine the entire manuscript.
>
> ***Q2: "how the definitions of stability and efficiency differ from those in previous studies"***
>
> **A2:**
> The only difference  between our definitions and those in previous studies is that our stability definition is slightly less restrictive.
> Specifically, we do not consider interval stability, i.e., $Z_A=AZ=\Theta(1)$, where $Z$ is the input of the LoRA layer. Instead, we focus on the stability of the final output of LoRA, $Z_B=BAZ=\Theta(1)$. Eq.(5) outlines the conditions that must be met by the initialization and learning rate when interval stability is excluded. A detailed discussion on interval stability is provided in Appendix B.2. Given that other reviewers have raised concerns regarding interval stability, we summarize the key points related to this topic in **Q3**.
>
> ***Q3: "interval stability"***
>
> **A3:**
> In this paper, stability is defined as $Z_B=BAZ=\Theta(1)$, where $Z$ represents the input to the LoRA layer. The condition $Z_B=\Theta(1)$ ensures the stability of LoRA's final output, while interval stability is defined as $Z_A=AZ=\Theta(1)$, which indicates the stability of LoRA's intermediate results. In Section 3, we present the solution set for stable and efficient learning without considering interval stability, as shown in Eq.(5) or as follows:
>
> $\gamma[\eta_A]+\gamma[\eta_B]=-1, \gamma[A_0] \leq \gamma[\eta_A]$ and $\gamma[B_0] \leq \gamma[\eta_B]$.
>
> When interval stability is considered (i.e., $Z_A=\Theta(1)$), an additional constraint is imposed: $\gamma[\eta_A]=-1$. Consequently, the solution set of the learning rate and initialization becomes Eq.(21) in Appendix B.2:
>
> $\gamma[A_0] \leq \gamma[\eta_A]=-1$ and $\gamma[B_0] \leq \gamma[\eta_B]=0$.
>
> Two important points should be noted here:
> 1. $\gamma[\eta_A]=-1$ and $\gamma[\eta_B]=0$ are the key findings of LoRA+, which suggest using a larger learning rate for the matrix $B$ in practical applications.
>
> 2. Regardless of the optimal value for $\gamma[\eta_A]$ and $\gamma[\eta_B]$, the conditions $\gamma[A_0] \leq \gamma[\eta_A]$ and $\gamma[B_0] \leq \gamma[\eta_B]$  must always be satisfied to ensure LoRA's stable and efficient learning. When both "$\leq$" become "=", the maximum robustness to the learning rate is achieved. Therefore, non-zero initialization can also enhance LoRA+'s robustness to the learning rate, as  shown in Fig.12 in the above link.

---

### Official Review · Reviewer_zK3d · 2025-03-14

**Overall Recommendation:** 3

**Summary:**

This paper investigates the impact of non-zero initialization in Low-Rank Adaptation (LoRA) fine-tuning, challenging the conventional practice of initializing one of the LoRA matrices (A or B) to zero. Through theoretical analysis and empirical validation, the authors demonstrate that simultaneously initializing A and B to non-zero values (Init[AB]) enhances LoRA’s robustness to suboptimal learning rates, particularly smaller ones, common due to learning rate decay. The study finds that while non-zero initialization introduces slight noise to the pre-trained model, it does not degrade fine-tuning performance as long as appropriate initialization variances are used. Extensive experiments across models and datasets confirm that non-zero initialization improves accuracy, stability, and convergence speed, making it practical for LoRA-based fine-tuning.

**Claims And Evidence:**

Well-supported claims:
1. Non-zero initialization improves LoRA’s robustness to suboptimal learning rates: This claim is supported by theoretical analysis and empirical proofs
2. Fine-tuning does not need to strictly start from the pre-trained model: Experiments and theoretical evidence provided
3. Non-zero initialization achieves superior performance compared to zero initialization, particularly at smaller learning rates: The heatmaps and performance tables demonstrate consistent improvements when using Init[AB] instead of Init[A], especially in low learning rate scenarios.

Claims that need more evidence:
1. Non-zero initialization improves performance in all cases: There is clearly a dependence on the learning rate for different tasks as can be seen in Tables 2,3
2. What are the limits on the variance of noise that can be used in the init[AB] case

**Essential References Not Discussed:**

The paper focuses on LoRA and the surrounding methods while not shedding light on other PEFT methods, such as BitFit [Zaken et al., 2022] and Adapters [Houlsby et al., 2019]. Adding these papers can help the reader understand the landscape better.

**Experimental Designs Or Analyses:**

Yes, I examined the soundness and validity of the experimental designs and analyses in the paper
1. The paper evaluates natural language understanding (GLUE benchmark) and natural language generation (commonsense & arithmetic reasoning), ensuring broad applicability.
2. The paper uses multiple model architectures: T5-Base (encoder-decoder) and Llama 3-8B (decoder-only transformer)
3. The study systematically varies the learning rate (η) and initialization variance (β), allowing a detailed exploration of their effects.
4. The heatmaps and accuracy tables provide clear evidence that non-zero initialization (Init[AB]) improves performance, particularly at lower learning rates.
5. The toy model experiment provides intuitive validation that non-zero initialization reduces sensitivity to learning rate choices.

There are areas where the text can improve with additional details
1. The reported accuracy differences (e.g., between Init[A] and Init[AB]) are sometimes small (e.g., ~1%).
2. No confidence intervals or standard deviations are provided
3. It’s unclear if different ranks or scaling factors would affect the relative performance of zero vs. non-zero initialization.
4. The initialization variance (β) is tested in discrete steps (e.g., {1, 2, 4, 8, 16}), but there’s no justification for why these values were chosen.
5. How does their method interact with version improvements of LoRA such as LoRA+, Asymmetric LoRA, etc.?

**Methods And Evaluation Criteria:**

The authors test their method with Llama-3 8B and T5 Models on the GLUE and arithmetic reasoning benchmarks. It would be interesting to check how their method works for other fine-tuning settings such as instruction tuning. Also, it is not clear how their method works with varients of LoRA such as Asymmetric LoRA, LoRA+, QLoRA, etc.,

**Other Comments Or Suggestions:**

See my comments above

**Other Strengths And Weaknesses:**

See my comments above

**Questions For Authors:**

See my questions in the analysis part.

**Relation To Broader Scientific Literature:**

This paper builds upon existing research in LoRA fine-tuning, neural network scaling, and weight initialization, challenging the conventional zero-initialization approach in LoRA. While prior work (Hu et al., 2022; Hayou et al., 2024a) focused on optimizing learning rates and rank selection, this study demonstrates that initializing both LoRA matrices (A and B) to non-zero values (Init[AB]) enhances robustness to suboptimal learning rates. Applying infinite-width scaling theory formalizes conditions for stable and efficient fine-tuning, extending insights from Kaiming initialization (He et al., 2015) to LoRA. Unlike recent empirical methods that use quantization errors (LoftQ), SVD (PISSA), or gradient-based initialization (LoRA-GA), this paper provides a theoretical foundation for non-zero initialization. It validates it with experiments across T5-Base, Llama 3-8B, and multiple benchmarks. These findings refine LoRA fine-tuning dynamics without additional computational cost, offering a practical and theoretically justified improvement.

**Theoretical Claims:**

No

---

> ### Author Rebuttal · Authors · 2025-03-31
>
> **Hi Reviewer zK3d:**
>
> Thank you for your detailed and insightful comments. Below, we provide responses to each point individually. Additional experimental results can be found in **https://anonymous.4open.science/r/nzlora_rebuttal-7D3E**.
>
> ***Q1: "accuracy's dependence on the learning rate in Tables 2,3, and accuracy differences are sometimes small"***
>
> **R1:**
> Our analysis reveals that non-zero initialization can reduce the adverse effects of suboptimal learning rates on LoRA performance. This effect is particularly evident when the learning rate is below its optimal value. However, when the learning rate approaches its optimal value, the performance improvement from non-zero initialization becomes less significant.
>
> ***Q2: "no justification for why $\beta\in\\{1,2,4,8,16\\}$"***
>
> **R2:**
> In our experiments, we set the initialization variance of matrices $A$ and $B$ as $\delta_A^2=\delta_B^2=(\beta \delta_k)^2$, where $\delta_k^2=1/n$ is the variance used in Kaiming initialization (the default setting of LoRA). Notably, $\beta$ does not strictly represent variance, but rather a scaling factor applied to $\delta_k$.
>
> Our analysis indicates that robustness improves as $\gamma[A_0]$ and $\gamma[B_0]$ approach −1/2. To explore this, we begin with standard Kaiming initialization ($\beta=1$, corresponding to $\gamma[A_0]=\gamma[B_0]=−1$) and systematically increase the variance $(\beta\in\{2,4,8,16\})$ to study its effects. Our experimental results (Figs.4 and 6 in the original paper) confirm that, within a certain range, increasing the initialization variance enhances robustness to learning rate variations and leads to better accuracy.
>
>
> ***Q3: "limits on the variance in Init[AB]"***
>
> **R3:**
> The theoretical limits of initialization variance are $\gamma[A_0] \leq -1/2$ and $\gamma[B_0] \leq -1/2$. However, this condition only describes the asymptotic behavior of the initialization variance as $n \to \infty$, rather than providing a specific value. To further investigate this, we performed ablation experiments on the variance limits of LLaMA 3-8B and T5-base models. As shown in Fig.16, these limits vary across models or datasets (e.g. Init[AB]-Init[A] with $\beta=4$ is generally less than 0 in the Commonsense reasoning task). However, the variance associated with Kaiming initialization (i.e., $\beta = 1$) is generally effective, yielding near-optimal accuracy.
>
>
> ***Q4: "different ranks or scaling fators"***
>
> **R4:**
> We conducted ablation experiments with varying ranks and scaling factors. As shown in in Fig.14 in the above link, adjusting these hyperparameters does not affect the improvement gained through non-zero initialization.
>
>
> ***Q5: "different fine-tuning settings"***
>
> **R5:**
> Following your suggestion, we conducted experiments using the an instruction tuning dataset, databricks-dolly-15k, and evaluated its performance on the MMLU task. As shown in Fig.15, non-zero initialization enhances LoRA's robustness to small learning rates in the instruction tuning task, thereby improving accuracy. Notably, LLama 3-8B exhibits limited accuracy on MMLU, and thus the improvement due to non-zero initialization is less pronounced. However, the trend is still observable.
>
>
> ***Q6: "LoRA variants"***
>
> **R6:**
> To address this question, we evaluated the impact of non-zero initialization on LoRA+ (using larger learning rates for matrices $B$) and HydraLoRA [1], an asymmetric LoRA variant (using one matrix $A$ with multiple matrices $B$). The results are presented in Figs.12-13 in the above link.
> 1. We tested LoRA+ on GLUE and Arithmetic reasoning tasks. The results show that appropriately increasing the learning rate of $B$ can indeed improve the model accuracy. Most importantly, for the same learning rate, non-zero initialization significantly enhances the accuracy of LoRA+.
> 2. We tested HydraLoRA on Arithmetic reasoning tasks. To ensure that the non-zero initialized $AB$ terms could be subtracted from the pre-trained weights, we use the same initialization for different $B$ matrices within a HydraLoRA layer. As shown in Fig.13, non-zero initialization also improves the robustness of HydraLoRA to the learning rate.
>
> [1] HydraLoRA: An Asymmetric LoRA Architecture for Efficient Fine-Tuning, NeurIPS 2024.
>
>
> ***Q7: "standard deviations"***
>
> **R7:**
> The standard deviation for the GLUE dataset ranges from 0.01 to 0.4, while for commonsense and Arithmetic reasoning tasks, it spans from 0.2 to 0.4. In our experiments, smaller learning rates tend to converge less effectively and exhibit higher standard deviations. However, this effect is minimal compared to the performance gains achieved by non-zero initialization. We will include the full standard deviation results in the revised paper.
>
> ***Q8: "surrounding methods such as BitFit and Adapters"***
>
> **R8:**
> Thank you for your suggestion. We will add a discussion of relevant PEFT methods in the revised paper.

---

### Official Review · Reviewer_uk4U · 2025-03-15

**Overall Recommendation:** 4

**Summary:**

This paper considers scaling of hyperparameters for LoRA finetuning from an infinite width perspective following [1, 2]. The key difference compared to past works is that a non-zero random initialization of both the B and A adapter matrices is considered. The initialization can  optionally be subtracted from the pretrained weights to ensure the overall layer starts at the pretrained weights. The key observations are that the new initialization scheme allows "robustness" in addition to previous desiderata such as stability and efficiency. Informally robustness refers to a lack of sensitivity in the scaling of certain quantities to learning rate hyperparameters. The authors demonstrate on a variety of tasks the superior performance of the new scheme and improved robustness to suboptimal learning rates.

[1] LoRA+: Efficient Low Rank Adaptation of Large Models - Soufiane Hayou, Nikhil Ghosh, Bin Yu
[2] The Impact of Initialization on LoRA Finetuning Dynamics - Soufiane Hayou, Nikhil Ghosh, Bin Yu

**Claims And Evidence:**

Yes the evidence is clear and convincing.

**Essential References Not Discussed:**

None.

**Experimental Designs Or Analyses:**

Yes the experimental designs are solid.

**Methods And Evaluation Criteria:**

Yes the methods and evaluation criteria make sense.

**Other Comments Or Suggestions:**

typos:
In Section B.2
Appendix Hu et al. reference is incorrect.
Last line of Eq. (20) should be Z_A^{t-1} not Z_B^{t-1}.

The presentation in Sections 3.2 and 3.3 are a bit hard to parse at first as is the definition and intent of "robustness". I don't think it is about perturbing \gamma[\eta] (which doesn't make much sense) but really perturbing \eta and in certain scalings the perturbation has a dominant quadratic dependence on \eta.

Also in the grid sweeps the optimum is in the top right corner. Can you extend beyond that to check that increasing the hyperparameters further does not bring improvements?

**Other Strengths And Weaknesses:**

The strength of this paper is that it expands the practical consideration of initializations for LoRA and offers evidence for the superiority of a new initialization. Empirically this initialization appears to be non-trivially better than the standard practice and is trivial to implement.

**Questions For Authors:**

Can the analysis say anything useful about PiSSa?

To clarify \eta_A = \eta_B is needed for maximum robustness? Also in this case internal stability is not achieved?

If we use LoRA+ and non-zero initialization will we do even better?

Using Init[AB] can reduce learning rate sensitivity, but it will increase initialization variance sensitivity compared to init[A]?

**Relation To Broader Scientific Literature:**

The paper is important for understanding the optimal setting of hyperparameters for LoRA finetuning, a popular parameter efficient finetuning method. The paper characterizes the scaling of certain quantities in terms of width and imposes various desiderata for finetuning akin to a variety of works such as [1, 2, 3]. Importantly this work goes beyond previous works by considering a non-zero initialization of LoRA. In particular they show that finetuning can be successful even when the non-zero initialization is not subtracted from the pretrained weights as long as the initialization variance is not too large, demonstrating robustness of the finetuning procedure to a noisy initialization. Furthermore, the non-zero initialization has certain advantages relative to other initializations including decreased sensitivity to learning rate hyperparameters and improved empirical performance.

[1] LoRA+: Efficient Low Rank Adaptation of Large Models - Soufiane Hayou, Nikhil Ghosh, Bin Yu
[2] The Impact of Initialization on LoRA Finetuning Dynamics - Soufiane Hayou, Nikhil Ghosh, Bin Yu
[3] Feature Learning in Infinite-Width Neural Networks - Greg Yang, Edward J. Hu

**Theoretical Claims:**

Yes the proofs appear correct.

---

> ### Author Rebuttal · Authors · 2025-03-31
>
> **Hi Reviewer uk4U:**
>
> Thank you for your detailed and insightful comments. Below, we provide responses to each point individually. Additional experimental results can be found in **https://anonymous.4open.science/r/nzlora_rebuttal-7D3E**.
>
> ***Q1: "typos"***
>
> **R1:**
> Thanks again for catching these! All typos will be fixed in the revision.
>
> ***Q2: "perturbing $\gamma[\eta]$ or $\eta$"***
>
> **R2:**
> This question is essential for understanding our analysis. To improve clarity, we restate our infinite-width analysis as follows:
>
> 1. In this paper, we focus on the asymptotic behavior of the learning rate, $\gamma[\eta]$, as the network width $n \to \infty$, rather than its exact value. The $\gamma$-operator is defined such that $\eta = \Theta(n^{\gamma[\eta]}) \approx c \cdot n^{\gamma[\eta]}$, where $c > 0$ is a constant and lower-order terms are neglected.
> 2. As $n \to \infty$, the term $n^{\gamma[\eta]}$ dominates, making $\gamma[\eta]$ the key factor determining the asymptotic behavior of $\eta$. While the constant $c$ is important for exact values, it doesn't influence the asymptotic scaling behavior. Ignoring the influence of $c$, perturbations to $\gamma[\eta]$ and $\eta$ are effectively equivalent.
> 3. Thus, we focus on how perturbations to $\gamma[\eta]$ affect the fine-tuning dynamics, excluding the constant $c$ in $\Theta$.
> Note that we analyze $\gamma[\eta]$ to guide learning rate and initialization choices, not compute exact values (which depend on $c$).
>
> We appreciate the opportunity to clarify our analytical framework and will explicitly incorporate these refinements in the revised paper.
>
> ***Q3: "extend beyond the top right corner"***
>
> **R3:**
> Following your suggestion, we have expanded the upper right corner of the heatmap, and the updated results are presented in Fig.16 in the above link. The results show that increasing the hyperparameters further does not lead to a substantial improvement in accuracy.
>
>
> ***Q4: "insights about PiSSA"***
>
> **R4:**
> Our analysis suggests that PiSSA is more robust to variations in the learning rate, owing to its use of non-zero initialized LoRA, which is achieved through Truncated SVD on pre-trained weights. Fig.11 in the above link shows that a significant portion of the improvement in PiSSA's accuracy can be attributed to non-zero initialization. The remaining improvement is due to the fact that the initialization values derived from the pre-trained weights are more effective than random noise. A similar trend is observed in LoRA-GA, which employs gradients for non-zero initialization. Please refer to Fig.11 for further details.
>
> ***Q5: "clarify the need of \eta_A = \eta_B, and the internal stability"***
>
> **R5:**
> In fact, the condition $\eta_A = \eta_B$ is not necessary for achieving maximum robustness. The solution set in Eq. (5) indicates that stable and efficient learning can be achieved as long as $\gamma[\eta_A] + \gamma[\eta_B] = -1$, $\gamma[A_0] \leq \gamma[\eta_A]$, and $\gamma[B_0] \leq \gamma[\eta_B]$. Under this condition, when both "$\leq$" become "=", maximum robustness is attained. By default, we set $\eta_A = \eta_B$ since fine-tuning typically employs a uniform learning rate for all LoRA weights. However, achieving internal stability further requires $\gamma[\eta_A] = -1$ and $\gamma[\eta_B] = 0$, which are the core propositions of LoRA+. Due to space limitations, further details on internal stability can be found in **R3 from reviewer_Rpo6** or in the analysis presented in Appendix B.2.
>
> ***Q6: "LoRA+ with non-zero initialization"***
>
> **R6:**
> We integrated LoRA+ with non-zero initialization. As shown in Fig.12 in the above link, non-zero initialization also enhances the robustness of LoRA+ to variations in the learning rate, leading to improved model accuracy.
>
> ***Q7: "initialization variance sensitivity of Init[AB]"***
>
> **R7:**
> Let's first explain the meaning of each initialization method:
>
> Init[A]:$A_0\sim \mathcal{N}(0, \delta^2), B_0=0$,
>
> Init[AB]:$A_0\sim \mathcal{N}(0, \delta^2), B_0\sim \mathcal{N}(0, \delta^2)$, with
> $\frac{\alpha}{r}A_0B_0$ subtracted from the pre-trained weights.
>
> Init[AB+]: Same as Init[AB], but without the subtraction process.
>
> First, we emphasize that Init[A] itself exhibits sensitivity to variance. As shown in Eq.(6), stable and efficient learning requires the variance of $A_0$ to satisfy $\gamma[A_0] \leq -\frac{1}{2}$. In Init[AB], $B_0$ uses the same variance as $A_0$ and only needs to satisfy the same condition (i.e., $\gamma[B_0]\leq-1/2$). Therefore, Init[AB] does not introduce additional sensitivity to initialization variance but instead enhances the robustness of LoRA to $\eta_B$.
>
> Notably, if Init[AB+] is used, a larger initialization variance results in greater noise, which negatively impacts performance. However, this issue arises due to the absence of noise subtraction in Init[AB+], rather than a fundamental limitation of Init[AB].

---

### Decision · Program_Chairs · 2025-05-01

**Decision:**

Accept (poster)

**Comment:**

This paper received four effective reviews, and all of them are positive. Overall, the paper is of good quality and should be accepted.